# What Does It Take to Build a Performant Selective Classifier?

**Stephan Rabanser**[*]
Princeton University
rabanser@princeton.edu

**Nicolas Papernot**
University of Toronto & Vector Institute
nicolas.papernot@utoronto.ca

## Abstract

Selective classifiers improve model reliability by abstaining on inputs the model deems uncertain. However, few practical approaches achieve the gold-standard performance of a perfect-ordering oracle that accepts examples exactly in order of correctness. Our work formalizes this shortfall as the *selective-classification gap* and present the first finite-sample decomposition of this gap to five distinct sources of looseness: Bayes noise, approximation error, ranking error, statistical noise, and implementation- or shift-induced slack. Crucially, our analysis reveals that monotone post-hoc calibration—often believed to strengthen selective classifiers—has limited impact on closing this gap, since it rarely alters the model's underlying score ranking. Bridging the gap therefore requires scoring mechanisms that can effectively reorder predictions rather than merely rescale them. We validate our decomposition on synthetic two-moons data and on real-world vision and language benchmarks, isolating each error component through controlled experiments. Our results confirm that (i) Bayes noise and limited model capacity can account for substantial gaps, (ii) only richer, feature-aware calibrators meaningfully improve score ordering, and (iii) data shift introduces a separate slack that demands distributionally robust training. Together, our decomposition yields a quantitative error budget as well as actionable design guidelines that practitioners can use to build selective classifiers which approximate ideal oracle behavior more closely.

## 1 Introduction

In high-stakes applications like finance [Coenen et al., 2020], healthcare [Guan et al., 2020], and autonomous driving [Ghodsi et al., 2021], machine learning (ML) models are increasingly tasked with making decisions under uncertainty, where dependable predictions are critical. Selective classifiers [Chow, 1957, El-Yaniv et al., 2010] formalize the option to abstain on inputs deemed unreliable, reducing the risk of costly errors by refusing to predict when uncertain. Their effectiveness depends on identifying which predictions to trust and which to defer. A common evaluation metric is the *accuracy–coverage* tradeoff, which quantifies how performance degrades as the model accepts a broader set of inputs. The benchmark is a hypothetical oracle that ranks inputs by their true likelihood of correctness, yielding a *perfect-ordering upper bound* [Geifman et al., 2019, Rabanser et al., 2023]. While some selective predictors approach this bound, others fall short—revealing persistent gaps and raising open questions about what properties of the learning setup truly govern selective performance.

Classical theory explains selective classification in two idealized regimes. In the *realizable* setting [El-Yaniv et al., 2010], where the data is noiseless and the true predictor lies within the hypothesis class, the model can asymptotically achieve the ideal accuracy–coverage curve. In the more general *agnostic* setting [Wiener and El-Yaniv, 2011], the classifier competes with the best-in-class predictor, but this benchmark may itself fall well below the oracle bound—and the theory does not isolate

---

[*]Work done while at the University of Toronto and the Vector Institute.

the source of the gap. Yet in practice, we never operate in such asymptotic or idealized conditions: models are often misspecified, the data used for training and evaluation are finite, and asymptotic guarantees offer little actionable insight. As a result, even the strongest formal guarantees provide limited guidance, which leaves practitioners with the following question:

> *For my finite model on finite data, what aspects of the learning setup will actually move my trade-off curve closer to the perfect-ordering upper bound?*

To answer this question, we re-frame selective performance around the *selective classification gap* $\Delta(c)$: the mismatch between a model's accuracy–coverage curve and the oracle bound for all coverage levels $c$ (see Figure 1). Our work shows that this gap admits a finite-sample decomposition:

$$\widehat{\Delta}(c) \;\leq\; \underbrace{\varepsilon_{\mathrm{Bayes}}(c)}_{\text{irreducible}} + \underbrace{\varepsilon_{\mathrm{approx}}(c)}_{\text{capacity}} + \underbrace{\varepsilon_{\mathrm{rank}}(c)}_{\text{ranking}} + \underbrace{\varepsilon_{\mathrm{stat}}(c)}_{\text{data}} + \underbrace{\varepsilon_{\mathrm{misc}}(c)}_{\text{optimization \& shift}} \;, \qquad \forall c \in (0,1]. \quad (1)$$

Each term corresponds to a distinct—and often *measurable*—source of looseness. The first term, $\varepsilon_{\mathrm{Bayes}}(c)$, reflects irreducible uncertainty: if the true label is inherently unpredictable from the input (e.g., due to label noise), even a perfect classifier must abstain on some examples. Next, $\varepsilon_{\mathrm{approx}}(c)$ captures limits of the model class: if the function class is too weak to approximate the Bayes-optimal decision rule, the gap widens. The third term, $\varepsilon_{\mathrm{rank}}(c)$, quantifies the model's failure to correctly order inputs by their likelihood of correctness—typically due to poor confidence estimation or miscalibration. The statistical term $\varepsilon_{\mathrm{stat}}(c)$ accounts for finite-sample fluctuations that affect both learning and evaluation. Finally, $\varepsilon_{\mathrm{misc}}(c)$ aggregates practical imperfections, such as optimization error or test-time distribution shift. Equation (1) thus provides a coverage-uniform "error budget" that transforms the qualitative question posed earlier into a concrete quantitative diagnosis.

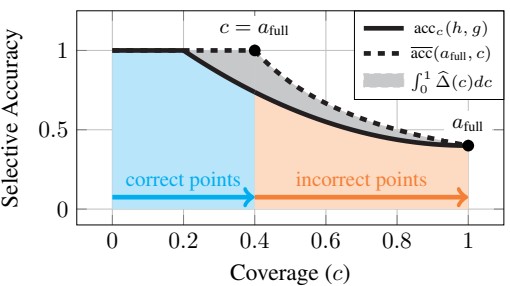

Figure 1: **Visualization of the selective classification gap** $\Delta(c)$**.** The dashed curve is the oracle frontier $\overline{\mathrm{acc}}(a_{\mathrm{full}}, c)$ under which coverage levels left of $c = a_{\mathrm{full}}$ (blue) accept all correct predictions first, and rank incorrect predictions last (orange). This constitutes the ideal behavior of a selective predictor. On the other hand, the solid curve shows the realized selective accuracy $\mathrm{acc}_c(h, g)$. The mismatch between $\overline{\mathrm{acc}}(a_{\mathrm{full}}, c)$ and $\mathrm{acc}_c(h, g)$ at coverage $c$ is the gap $\Delta(c)$; the gray shaded area visualizes this gap over the full coverage spectrum.

Two key insights, developed further in later sections, are worth previewing. First, we show that monotone post-hoc calibration—a technique that is often thought to improve selective prediction performance—only possess a limited ability of reducing the *ranking* term $\varepsilon_{\mathrm{rank}}(c)$. In contrast, methods that directly yield improved uncertainty scores by leveraging richer feature representations or aggregating diverse model perspectives dominate post-hoc calibration methods. Second, Equation (1) serves as an *error budget* that identifies cost-effective levers leading to actionable recommendations for practitioners: (i) use additional or repeated labels and noise-robust losses to reduce $\varepsilon_{\mathrm{Bayes}}$; (ii) increase capacity or distill from a more expressive teacher to shrink $\varepsilon_{\mathrm{approx}}$; (iii) enlarge validation data to lower $\varepsilon_{\mathrm{stat}}$; and (iv) apply domain adaptation or importance weighting to address $\varepsilon_{\mathrm{misc}}$.

**Contributions.** We summarize our main contributions below:

- **Problem formulation.** We recast selective prediction in terms of a *coverage-uniform selective classification gap*—the key quantity to minimize to approach perfect selective prediction. This framing unifies prior work and highlights which failure modes dominate at each coverage level.

- **Theoretical analysis.** We present the first *finite-sample decomposition* of the selective classification gap (Equation (1)), dividing it into five terms: Bayes, approximation, ranking, statistical, and miscellaneous errors. Our analysis further shows that *monotone calibration is ineffective at reducing the gap*, motivating the use of methods that can change the ranking more flexibly.

- **Empirical validation.** Our synthetic and real-world experiments confirm the decomposition: Bayes noise and capacity limits drive large gaps; temperature scaling improves calibration but not ranking; and shift-aware methods remain essential under distribution shift. *These results clarify which factors matter most and how to target them effectively in practice.*

## 2 Background & Related Work on Selective Classification

Selective classification extends the standard supervised classification framework as follows:

> **Definition 1** (Selective Classifier [Chow, 1957, El-Yaniv et al., 2010]). A selective classifier is a pair $(h, g)$, where $h : \mathcal{X} \to \mathcal{Y}$ is a classifier over covariates $\mathcal{X} = \mathbb{R}^D$ and labels $\mathcal{Y} = \{1, \ldots, K\}$, and $g : \mathcal{X} \times (\mathcal{X} \to \mathcal{Y}) \to \mathbb{R}$ is a selection function that assigns a confidence score. Given a threshold $\tau \in \mathbb{R}$, the model abstains when the score falls below the threshold:
> $$(h, g)(x) = \begin{cases} h(x) & \text{if } g(x, h) \geq \tau \\ \perp & \text{otherwise} \end{cases}. \tag{2}$$

Intuitively, a selective classifier predicts only when confident. The selection score $g(x, h)$ determines whether to accept or abstain: if $g(x, h) \geq \tau$, the model outputs $h(x)$; otherwise, it returns $\perp$.

Many prior works have developed selective classification methods for training competitive pairs $(h, g)$. A popular method is *Maximum Softmax Probability* (MSP) [Hendrycks and Gimpel, 2017, Geifman and El-Yaniv, 2017], which uses classifier confidence as the selection score. To improve calibration and reduce predictive variance, ensembling approaches have been explored: *Deep Ensembles* (DE) [Lakshminarayanan et al., 2017] train multiple models with different initializations, while *Selective Classification via Training Dynamics* (SCTD) [Rabanser et al., 2022] ensembles intermediate checkpoints. Other methods—such as *SelectiveNet* (SN)[Geifman and El-Yaniv, 2019], *Deep Gamblers* (DG)[Liu et al., 2019], and *Self-Adaptive Training* (SAT) [Huang et al., 2020]—alter the model architecture or loss function ensuring that prediction and rejection are learned jointly.

The efficacy of a selective classifier is evaluated using the empirical accuracy-coverage tradeoff.

> **Definition 2** (Empirical Accuracy–Coverage Tradeoff). Let $D = \{(x_i, y_i)\}_{i=1}^{N}$ be a dataset. For a selective classifier $(h, g)$ and threshold $\tau$, define
> $$\hat{\xi}_{h,g}(\tau) = \frac{1}{N} \left| \{ i : g(x_i, h) \geq \tau \} \right|, \tag{3}$$
> $$\hat{\alpha}_{h,g}(\tau) = \begin{cases} \dfrac{\left| \{ i : h(x_i) = y_i \text{ and } g(x_i, h) \geq \tau \} \right|}{\left| \{ i : g(x_i, h) \geq \tau \} \right|}, & \text{if } \hat{\xi}_{h,g}(\tau) > 0, \\ 0, & \text{if } \hat{\xi}_{h,g}(\tau) = 0. \end{cases} \tag{4}$$
> The pair $(\hat{\xi}, \hat{\alpha})$ as $\tau$ varies is the empirical accuracy–coverage curve.

The score $g(x, h)$ therefore induces a total order over $D$: $x_1$ is accepted before $x_2$ if $g(x_1, h) > g(x_2, h)$. This ordering governs which inputs are retained as coverage decreases. Effective strategies aim to maximize $\hat{\alpha}$ at each coverage level $\hat{\xi}$, often trading off accuracy and coverage.

**Accuracy–coverage tradeoff evaluation.** The accuracy–coverage tradeoff is often summarized by the area under the accuracy–coverage curve (AUACC), integrating selective accuracy over all coverage levels [Traub et al., 2024]. However, Geifman et al. [2019] show that AUACC favors models already accurate at full coverage. To address this issue, Geifman et al. [2019] and Rabanser et al. [2023] propose oracle-based bounds, which become loose at low utility [Galil et al., 2023]. To avoid accuracy bias, Galil et al. [2023] and Pugnana and Ruggieri [2023] recommend using the classifier's AUROC instead. But AUROC is not monotonic in AUACC [Cattelan and Silva, 2023, Ding et al., 2020], thus favoring methods tuned for AUROC over selective accuracy. Recently, Traub et al. [2024] introduced the Area Under the Goals-Reweighted Curve (AUGRC), which multiplies accuracy by coverage to mitigate bias toward low-coverage regions, while Mucsányi et al. [2024] provide a benchmark disentangling uncertainty sources for fairer comparison. These efforts refine *evaluation metrics*, whereas our work complements them by analyzing *what causes* selective performance gaps. Earlier work [El-Yaniv et al., 2010, Wiener and El-Yaniv, 2011] characterizes optimal selective classifiers in both realizable and agnostic regimes but focuses on existence rather than practical instantiation—unlike our finite-sample perspective.

# 3 Decomposing the Selective Classification Gap

We characterize the optimal performance achievable by a selective classifier given its full-coverage accuracy, establishing a reference against which all practical selective classifiers can be evaluated.

## 3.1 Oracle Bound and Selective Classification Gap

**Definition 3** (Perfect Ordering Upper Bound [Geifman et al., 2019, Rabanser et al., 2023]). Fix a base classifier $h$ whose *full-coverage* (standard) accuracy is $a_{\text{full}} := \Pr\big(h(X) = Y\big) \in [0, 1]$. For any desired coverage level $c \in (0, 1]$, the best selective accuracy—achieved by accepting the $c$-fraction of points with the *highest* posterior correctness $\Pr(h(X) = Y \mid X)$—is

$$\overline{\text{acc}}\big(a_{\text{full}}, c\big) = \begin{cases} 1, & 0 < c \leq a_{\text{full}}, \\ \dfrac{a_{\text{full}}}{c}, & a_{\text{full}} < c < 1. \end{cases} \tag{5}$$

Assuming no Bayes noise—that is, all errors are avoidable given perfect confidence—this piecewise curve (see Figure 1) traces the *oracle* accuracy–coverage frontier based on a perfect ranking of examples by correctness probability. Any real selective classifier falls below this bound—potentially far below, depending on its calibration, expressivity, and sensitivity to noise. To quantify how far a given classifier falls short of this ideal, we define the *selective classification gap*.

**Definition 4** (Selective Classification Gap). Let $(h, g)$ be a selective classifier with full-coverage accuracy $a_{\text{full}} = \Pr(h(X) = Y)$. For a coverage level $c \in (0, 1]$, let $\tau_c$ be the threshold satisfying $\Pr\big(g(X, h) \geq \tau_c\big) = c$. The *selective accuracy* at coverage $c$ is $\text{acc}_c(h, g) := \Pr\big(h(X) = Y \mid g(X, h) \geq \tau_c\big)$. The *selective classification gap* at coverage $c$ is then defined as the deviation from the perfect-ordering upper bound:

$$\Delta(c) := \overline{\text{acc}}\big(a_{\text{full}}, c\big) - \text{acc}_c(h, g). \tag{6}$$

This gap $\Delta(c)$ can be interpreted as the *excess selective risk* at a given coverage $c$. We note that integrating $\Delta(c)$ over the entire coverage spectrum, $\int_0^1 \Delta(c) dc$, is equivalent to the definition of the Excess-AURC (E-AURC) metric proposed by Geifman et al. [2019].

The function $\Delta(c)$ offers a coverage-resolved diagnostic of selective performance. A small gap indicates near-oracle behavior—accepting only examples it can confidently and correctly classify—while a large gap suggests limitations in estimating correctness or ranking examples reliably. Understanding the magnitude and shape of this gap is key to analyzing and improving selective classifiers.

## 3.2 Why Is the Upper Bound Loose?

The oracle bound in Definition 3 relies on two idealized assumptions: perfect prediction on all inputs and perfect ranking by the true correctness posterior. In practice, selective classifiers deviate in four principal ways, each corresponding to a term in our later decomposition ($\varepsilon_{\text{Bayes}}, \varepsilon_{\text{approx}}, \varepsilon_{\text{rank}}, \varepsilon_{\text{stat}}$):

1. **Bayes noise ($\varepsilon_{\text{Bayes}}$).** Even a Bayes-optimal rule errs on intrinsically ambiguous points (where $\max_y \Pr(Y = y \mid X) < 1$), unavoidable in real data [Devroye et al., 2013]. As coverage increases, the oracle must accept some of these noisy inputs, lowering the achievable accuracy.

2. **Approximation limits ($\varepsilon_{\text{approx}}$).** A learned model $h$ drawn from a restricted hypothesis class may misclassify inputs with high posterior confidence under the Bayes rule [Bishop, 2006]. This gap reduces full-coverage accuracy and limits selective performance.

3. **Ranking error ($\varepsilon_{\text{rank}}$).** Let $\eta_h(x) := \Pr\big(h(x) = Y \mid X = x\big)$ denote the true correctness posterior, i.e., the probability that the model's prediction is correct given the input. Ideally, the confidence score $g(X, h)$ should rank examples in decreasing order of $\eta_h(x)$—so that samples the model is likely to classify correctly (high $\eta_h(x)$, examples that are "easy") are accepted before those it is likely to misclassify (low $\eta_h(x)$, examples that are "hard"). When $g(X, h)$ fails to preserve this ordering, high-confidence errors and low-confidence corrects are interleaved, increasing the selective gap $\Delta(c)$.

4. **Statistical noise ($\varepsilon_{\text{stat}}$).** Estimating the threshold $\tau_c$ and selective accuracy from a finite validation set introduces randomness of order $\mathcal{O}(\sqrt{\log(1/\delta)/n})$. This follows from concentration bounds; see Shalev-Shwartz and Ben-David [2014] for standard applications in learning theory.

> **Takeaway.** The selective classification gap $\Delta(c)$ reflects a mix of irreducible noise, model misspecification, ranking errors, and sampling variability. Addressing each—via cleaner labels, stronger models, or improved ranking—can tighten selective prediction performance.

In the next subsection, we formalize this decomposition and provide a general bound on the total gap.

### 3.3   Formal Decomposition of the Gap

We now give a principled decomposition of the selective classification gap and provide a corresponding finite-sample upper bound. For clarity and notational simplicity, we treat the binary-label case $\mathcal{Y} = \{0, 1\}$; the multiclass extension follows by a standard one-vs-rest reduction.

**Notation.** Let $\eta(x) := \Pr(Y = 1 \mid X = x)$ be the Bayes posterior. For a fixed classifier $h : \mathcal{X} \to \mathcal{Y}$ define its (induced) correctness posterior

$$\eta_h(x) := \Pr(h(x) = Y \mid X = x) = \eta(x)\mathbb{I}_{\{h(x)=1\}} + (1 - \eta(x))\mathbb{I}_{\{h(x)=0\}}. \tag{7}$$

All expectations and probabilities are taken w.r.t. the true data distribution $\mathcal{D}$. Throughout let $g(x, h)$ be the confidence score. For a target coverage $c \in (0, 1]$ denote by

$$t_c \quad \text{s.t.} \quad \Pr(g(X, h) \geq t_c) = c \tag{8}$$

the *population threshold*, and write the *accepted region* $A_c := \{x : g(x, h) \geq t_c\}$. The oracle that attains the perfect-ordering bound accepts $A_c^\star := \{x : \eta_h(x) \text{ is among the largest } c\text{-fraction}\}$.

**Error terms.** We isolate the following sources of error affecting selective prediction performance:

$$\varepsilon_{\text{Bayes}}(c) := \mathbb{E}\Big[1 - \max\{\eta(X), 1 - \eta(X)\} \,\Big|\, X \in A_c\Big], \tag{9}$$

$$\varepsilon_{\text{approx}}(c) := \mathbb{E}\Big[\big|\eta_h(X) - \eta(X)\big| \,\Big|\, X \in A_c\Big], \tag{10}$$

$$\varepsilon_{\text{rank}}(c) := \mathbb{E}\big[\eta_h(X) \mid X \in A_c^\star\big] - \mathbb{E}\big[\eta_h(X) \mid X \in A_c\big] \quad (\geq 0), \tag{11}$$

$$\varepsilon_{\text{stat}}(c) := C\sqrt{\frac{\log(1/\delta)}{n}}, \tag{12}$$

where $n$ is the evaluation-set size, $\delta \in (0, 1)$ a confidence parameter, and $C > 0$ an absolute constant. Intuitively, $\varepsilon_{\text{Bayes}}$ is the irreducible label noise inside the accepted region; $\varepsilon_{\text{approx}}$ measures how far $h$ is from Bayes-optimal on the *selected* inputs; $\varepsilon_{\text{rank}}$ is a *ranking regret* measuring the accuracy loss due solely to picking the wrong $c$-fraction of samples; and $\varepsilon_{\text{stat}}$ captures the *sampling uncertainty* due to evaluating on a finite dataset. Note that we freeze the acceptance set $A_c$ defined by the current scoring function and ask how much worse the learned classifier $h$ is than the Bayes-optimal rule.

> **Remark** (Distance to a Perfect Ranker). A natural way to gauge how far the learned acceptance rule is from the oracle is the *mass mis-ordered*
>
> $$D_{\text{rank}}(c) := \Pr\big(X \in A_c^\star \setminus A_c\big) + \Pr\big(X \in A_c \setminus A_c^\star\big). \tag{13}$$
>
> It equals the total probability of examples that would have to be *swapped* between $A_c$ and $A_c^\star$ to recover perfect ordering. Hence $D_{\text{rank}}(c) = 0$ iff $A_c = A_c^\star$, in which case $\varepsilon_{\text{rank}}(c)$ also vanishes.

> **Theorem 1** (Selective Gap Bound). For a coverage level $c \in (0, 1]$ and a selective classifier $(h, g)$ the population gap obeys
>
> $$\Delta(c) = \overline{\text{acc}}(a_{\text{full}}, c) - \text{acc}_c(h, g) \leq \varepsilon_{\text{Bayes}}(c) + \varepsilon_{\text{approx}}(c) + \varepsilon_{\text{rank}}(c). \tag{14}$$

Let $\widehat{\Delta}(c)$ be the empirical gap on $n$ i.i.d. test points. Then, with probability at least $1 - \delta$,

$$\widehat{\Delta}(c) \leq \varepsilon_{\text{Bayes}}(c) + \varepsilon_{\text{approx}}(c) + \varepsilon_{\text{rank}}(c) + C\sqrt{\tfrac{\log(1/\delta)}{n}}. \tag{15}$$

**Proof.** Because $\text{acc}_c(h, g) = \mathbb{E}[\eta_h(X) \mid A_c]$, the gap decomposes as

$$\Delta(c) = \underbrace{\mathbb{E}[\eta_h \mid A_c^\star] - \mathbb{E}[\eta_h \mid A_c]}_{\varepsilon_{\text{rank}}(c)} + \underbrace{\mathbb{E}[\eta_h - \mathbb{I}_{\{h=Y\}} \mid A_c]}_{\varepsilon_{\text{approx}}(c)} + \underbrace{\mathbb{E}[1 - \max\{\eta, 1 - \eta\} \mid A_c]}_{\varepsilon_{\text{Bayes}}(c)}.$$

This yields the population bound (14). For each expectation in the decomposition apply Hoeffding's inequality, a union bound over the three terms gives, with probability $1 - \delta$, $|\widehat{\Delta}(c) - \Delta(c)| \leq C\sqrt{\log(1/\delta)/n}$. Adding this deviation to (14) establishes (15). See Appendix B.1 for an extended proof with detailed intermediate steps. $\square$

**A single design choice can shrink multiple error terms.** We note that the individual error terms from the decomposition in Equation (15) can still interact with each other. For example, when the confidence score is the maximum softmax probability (MSP), a better approximation of the true conditional $\eta$ not only lowers the *approximation* term $\varepsilon_{\text{approx}}(c)$ but also tends to align MSP more closely with $\eta_h$, thereby *indirectly reducing* the ranking error $\varepsilon_{\text{rank}}(c)$. Conversely, a non-monotone calibration head can reduce $\varepsilon_{\text{rank}}(c)$ without improving $\varepsilon_{\text{approx}}(c)$.

### 3.4 Calibration and Its (Limited) Effect on the Gap

As shown in Theorem 1, the selective classification gap includes a *ranking error* term $\varepsilon_{\text{rank}}(c)$, which captures misalignment between the confidence score and true correctness. Model calibration [Niculescu-Mizil and Caruana, 2005]—widely used to reduce over- or underconfidence—is often assumed to improve this alignment by transforming scores to better reflect correctness likelihood. Yet its effect on selective performance remains ambiguous and context-dependent. Prior work has reached conflicting conclusions: Zhu et al. [2022] argue that calibration may degrade abstention behavior, while Galil et al. [2023] find that temperature scaling can improve selective prediction in practice. We show that the impact on the gap depends critically on the *type* of calibration method used and its influence on the induced ranking. We begin by recalling the formal definition of calibration.

**Definition 5** (Perfect Calibration). For each input $x$ let a model produce a predicted label $\hat{y}(x)$ and an associated confidence score $s(x) \in [0, 1]$. We say the model is *perfectly calibrated* if

$$\Pr\big(Y = \hat{y}(X) \mid s(X) = t\big) = t \qquad \text{for every confidence level } t \in [0, 1]. \tag{16}$$

Practical estimators approximate (16) via a post-hoc map $\phi$ such that $\tilde{s}(x) = \phi(s(x))$ approaches prefect calibration. *Expected Calibration Error (ECE)* [Naeini et al., 2015] quantifies this closeness:

$$\text{ECE} = \sum_{b=1}^{B} \frac{|I_b|}{n} \left| \frac{1}{|I_b|} \sum_{i \in I_b} \mathbb{I}\{\hat{y}(x_i) = y_i\} - \frac{1}{|I_b|} \sum_{i \in I_b} \tilde{s}(x_i) \right|, \tag{17}$$

where $I_b$ is the set of indices in bin $b$, $n$ is the total number of examples, and $B$ is the number of bins.

**Monotone score-level calibration leaves the gap intact.** Isotonic regression [Zadrozny and Elkan, 2002] and histogram binning [Zadrozny and Elkan, 2001] fit a *monotone* $\phi: [0, 1] \to [0, 1]$ that preserves score ordering. Because monotone maps preserve ordering, the acceptance set $A_c = \{x : \tilde{s}(x) \geq \tau_c\}$ is identical to the one obtained from $s(x)$; hence the selective accuracy $\text{acc}_c(h, g)$ and the gap $\Delta(c) = \overline{\text{acc}}(a_{\text{full}}, c) - \text{acc}_c(h, g)$ are *unchanged*. Monotone calibration thereby reduces the approximation error $\varepsilon_{\text{approx}}(c)$ in Section 3.3 but leaves the ranking error $\varepsilon_{\text{rank}}(c)$ untouched.

**The effect of temperature scaling on the SC gap.** Temperature scaling, the most widely used post-hoc calibration technique, divides every logit vector $z(x) \in \mathbb{R}^K$ by a scalar $T > 0$,

$$p_j^{(T)}(x) = \frac{\exp(z_j(x)/T)}{\sum_k \exp(z_k(x)/T)}. \tag{18}$$

While this operation leads to a *monotone* rescaling of the *logits*, it can lead to a *non-monotone* rescaling of the *softmax probabilities*. Since the softmax function is non-linear with respect to the temperature parameter, temperature scaling can therefore change the ranking of samples by confidence. This re-ranking can lead to small but empirically validated improvements in selective classification performance, as measured by metrics like AUROC [Galil et al., 2023, Cattelan and Silva, 2023]. However, the magnitude of this effect is inherently limited (see Appendix B.2 for an extended discussion). While temperature scaling can refine the ordering, it does not fundamentally alter the underlying quality of the model's uncertainty estimates.

**Moving the gap requires non-monotone scoring.** To reduce the selective classification gap $\Delta(c)$, it is not enough to calibrate scores post-hoc using monotone mappings. One must actively change the ranking of accepted examples to better reflect their true likelihood of correctness. Achieving this typically requires non-monotone scoring mechanisms that incorporate richer, instance-specific information—such as deep ensembles (DE), self-adaptive training (SAT), or learned correctness heads $g_\psi(x)$ that map hidden representations to confidence estimates. These approaches leverage model diversity, stochasticity, or internal feature structure to distinguish samples that would otherwise receive identical or wrongly ordered confidence values under standard softmax outputs.

**Why binning and vector scaling should not be used.** Histogram binning [Naeini et al., 2015] and vector/Dirichlet scaling [Kull et al., 2019]—while widely to improve calibration—are poorly suited for selective classification. Histogram binning quantizes scores into a small number of bins, mapping wide score intervals to the same value and destroying within-bin ordering, which leads to effectively random selection among tied examples. Vector and Dirichlet scaling are post-hoc calibration methods that generalize temperature scaling by learning class-specific transformations of logits—vector scaling applies a linear transformation, while Dirichlet scaling interprets the logits as parameters of a Dirichlet distribution to better model uncertainty. Recent work by Le-Coz et al. [2024] shows that histogram binning and vector/Dirichlet scaling consistently degrade AUROC in selective classification. These results underscore our central claim: improving calibration does not guarantee better ranking. Reducing the selective classification gap requires score functions that explicitly learn to separate easy from hard examples, not just to produce better-calibrated probabilities.

**Loss prediction as a multicalibration litmus test.** A complementary view on how calibration connects to ranking ability arises from the notion of *multicalibration* [Hébert-Johnson et al., 2018], which requires that a model's confidence be calibrated not only overall but also across many subgroups of inputs. Recent work by Gollakota et al. [2025] shows that achieving strong multicalibration is equivalent to learning an accurate predictor of one's own loss—that is, training an auxiliary model to estimate, for each input, the probability that the base predictor will be correct. Viewed this way, reliability becomes a self-forecasting problem: if a model (or an auxiliary head) can successfully predict its own 0–1 loss, then its confidence scores must already be well aligned with correctness, leaving little residual ranking error. We formalize this equivalence in Appendix E and show, both theoretically and empirically, that the degree to which a model's loss can be predicted corresponds directly to the magnitude of the ranking-error term $\varepsilon_{\mathrm{rank}}(c)$. In short, when no auxiliary predictor can outperform the model's own confidence scores at identifying its mistakes, the model is effectively multicalibrated and near the oracle frontier; conversely, any nontrivial loss-prediction advantage exposes where—and by how much—its internal ranking deviates from perfect ordering.

> **Takeaway.** While post-hoc calibration with temperature scaling can provide modest improvements to ranking, it is not sufficient to close the SC gap. Substantially reducing the ranking error ($\varepsilon_{\mathrm{rank}}$) requires more powerful scoring methods that actively re-rank examples based on richer information, such as feature-aware heads, ensembles, or non-monotone transformations.

## 3.5 Additional Practical Sources of Looseness

The decomposition in Theorem 1 captures the *intrinsic* sources of error—Bayes noise, approximation limits, ranking error, and sampling slack—forming a principled bound that holds even under perfect optimization, infinite data, and i.i.d. testing. In practical deployments, however, additional imperfections can inflate the empirical gap $\widehat{\Delta}(c)$. These stem from implementation details, scoring granularity, and distribution shift—not fundamental limits, but contingent slack terms reducible through better engineering. We summarize them below under a single *residual slack* term $\varepsilon_{\mathrm{misc}}(c)$.

1. **Optimization error $\varepsilon_{\text{opt}}$.** In practice, gradient-based solvers rarely attain the empirical risk minimizer. If $L(\theta)$ denotes the end-to-end training objective—encompassing model architecture, loss (e.g. cross-entropy), and training data—and $\hat{\theta}$ its final iterate, then $\varepsilon_{\text{opt}} = L(\hat{\theta}) - \min_\theta L(\theta)$, which—via standard surrogate-to-0/1 calibration bounds—translates into a nonzero selective-accuracy loss that persists even under infinite data.

2. **Distribution shift $\varepsilon_{\text{shift}}(c)$.** When the test distribution $p_{\text{test}}$ deviates from the training distribution $p_{\text{train}}$, both calibration and ranking typically degrade. In particular, for a hypothesis class $\mathcal{H}$, the gap due to shift can be bounded by an *Integral Probability Metric (IPM)* [Müller, 1997]:

$$\varepsilon_{\text{shift}}(c) \leq \text{IPM}_{\mathcal{H}}(p_{\text{train}}, p_{\text{test}}) := \sup_{f \in \mathcal{H}} |\mathbb{E}_{p_{\text{train}}}[f] - \mathbb{E}_{p_{\text{test}}}[f]|. \tag{19}$$

Hence, larger shifts in distribution (relative to $\mathcal{H}$) lead to wider selective classification gaps.

**Residual slack.** The dominant practical sources of looseness are optimization error and distribution shift, summarized by $\varepsilon_{\text{misc}}(c) := \varepsilon_{\text{opt}} + \varepsilon_{\text{shift}}(c)$. These two terms capture the main drivers of residual deviation between the theoretical and empirical gaps. For completeness, we discuss additional minor contributors such as threshold-selection noise or score quantization in Appendix B.3. Together, these effects make the bound in Equation (20) *sufficient*, not merely necessary, for explaining all observed looseness in practical selective classifiers, yielding the streamlined high-probability bound.

$$\widehat{\Delta}(c) \leq \underbrace{\varepsilon_{\text{Bayes}}(c) + \varepsilon_{\text{approx}}(c) + \varepsilon_{\text{rank}}(c) + \varepsilon_{\text{stat}}(c)}_{\text{intrinsic}} + \varepsilon_{\text{misc}}(c). \tag{20}$$

> **Takeaway.** Only $\varepsilon_{\text{Bayes}}$ reflects irreducible uncertainty; the other intrinsic terms—$\varepsilon_{\text{approx}}$, $\varepsilon_{\text{rank}}$, and $\varepsilon_{\text{stat}}$—can be reduced with better models, calibration, and data. The *miscellaneous slack* $\varepsilon_{\text{misc}}$ highlights optimization and shift-robustness as levers for closing the gap to the oracle.

## 4 Empirical Results

Our experimental study is organized around three guiding questions that reflect the theoretical decomposition in Section 3. Unless otherwise specified, all results are averaged over 5 random seeds.

### 4.1 Q1: How do Bayes error and approximation error shape the gap?

*Setup.* We conduct both synthetic and real-world experiments. For our synthetic results, which give us precise control over the data generation process, we simulate two sources of intrinsic difficulty on the two-moons dataset: (i) **noise** $\sigma \in \{0.1, 0.33, 0.66, 1.5\}$ controls how much the two moons expand into each other; and (ii) **model capacity**, varied from logistic regression (low capacity) to a shallow MLP (high capacity). For our real-world experiments we tackle the analysis similarly: for (a) we evaluate a trained CIFAR-10 model on the CIFAR-10N/100N [Wei et al., 2022] datasets to assess which data points have large labeling disagreement; and for (b) we vary the model architecture across a simple CNN (details in Appendix D.3), a ResNet-18 [He et al., 2016], and a WideResNet-50 [Zagoruyko and Komodakis, 2016] on CIFAR-100 [Krizhevsky et al., 2009] and StanfordCars [Krause et al., 2013]. For each setting, we compute the Excess-AURC (E-AURC) [Geifman et al., 2019] by integrating the empirical gap $\widehat{\Delta}(c)$ across all coverage levels.

*Findings.* In terms of approximation error, Figure 2 demonstrates that limited model capacity leads to larger gaps, while more expressive models yield tighter alignment with the perfect-ordering bound. This suggests that approximation error is a key driver of looseness. In terms of Bayes error, Figure 3 shows that increasing label noise consistently lowers the accuracy–coverage curve, indicating that Bayes error introduces an irreducible component to the gap. These results validate the canonical bound (Equation (15)): large Bayes or approximation error can explain substantial looseness.

### 4.2 Q2: When—and what kind of—calibration helps?

*Setup.* We study the same three model classes as before on CIFAR-100: a lightweight CNN, a ResNet-18, and a WideResNet-50. On each backbone we evaluate the following confidence–scoring variants: (i) maximum softmax probability (MSP) [Hendrycks and Gimpel, 2017];

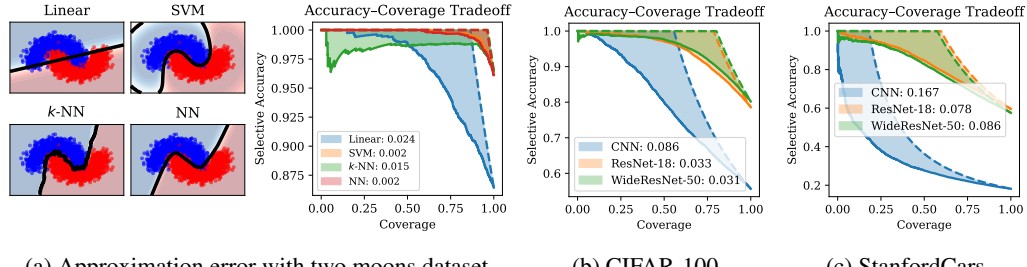

(a) Approximation error with two moons dataset.  (b) CIFAR-100  (c) StanfordCars

Figure 2: **Experiments on approximation error**. We find that approximation error is a major contributor to the gap. (a) We show the two moons dataset fitted with models of different degrees of expressiveness as well as the corresponding accuracy-coverage tradeoffs. (b) + (c) Accuracy-coverage tradeoffs for various model architectures on CIFAR-100 and StanfordCars, respectively.

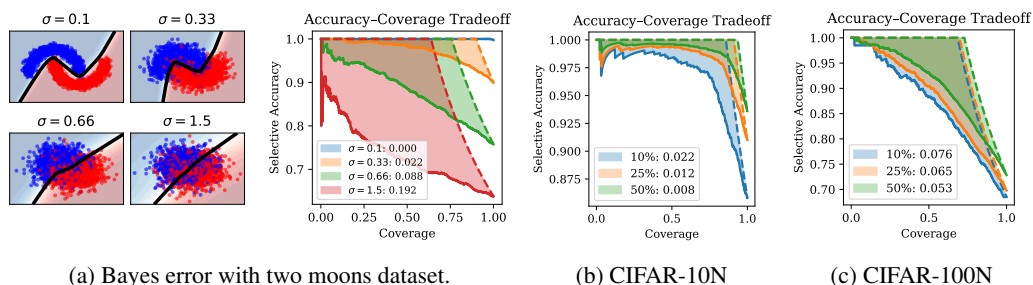

(a) Bayes error with two moons dataset.  (b) CIFAR-10N  (c) CIFAR-100N

Figure 3: **Experiments on Bayes error**. We find that irreducible noise significantly contributes to the gap. (a) We show the two moons dataset with varying degrees of noise $\sigma \in \{0.1, 0.33, 0.66, 1.5\}$ as well as the corresponding accuracy-coverage tradeoffs. (b) + (c) Accuracy-coverage tradeoffs for the 10% (blue), 25% (orange), and 50% (green) most noisy images in CIFAR-10N/100N, respectively.

(ii) a temperature-scaled softmax (monotone probability calibration, TEMP) [Guo et al., 2017]; (iii) self-adaptive training (SAT) [Huang et al., 2020], which implicitly calibrates by relabelling uncertain samples during training; and (iv) deep ensembles (DE) [Lakshminarayanan et al., 2017] of five independently initialised networks (non-monotone aggregation; improves ranking via variance). Our inclusion of the MSP baseline is motivated by the large-scale study of Jaeger et al. [2023], who find that MSP, while simple and easy to implement, is often hard-to-beat in practice. For each score we report (a) the weighted Expected Calibration Error (ECE); and (b) the Excess-AURC (E-AURC) [Geifman et al., 2019] metric measuring selective prediction performance.

*Findings.* We summarize our findings in Table 1. While temperature scaling (TEMP) consistently improves ECE across model classes relative to MSP, it leaves the selective classification gap largely unchanged—highlighting the limitations of monotone calibration. In contrast, SAT slightly improves both ECE and gap by perturbing rankings through relabeling, while deep ensembles (DE) achieve the largest gap reductions by explicitly reordering predictions via averaging. These trends confirm that only methods capable of re-ranking—implicitly (SAT) or explicitly (DE)—can meaningfully improve selective performance. Consistent with this, we find that only SAT and DE models reliably predict their own loss, reinforcing their stronger alignment with correctness. See Appendix E.5 for details.

### 4.3 Q3: How does the gap evolve under distribution shift?

*Setup.* As in Q1, we explore this question using both synthetic and real-world distribution shifts. For synthetic experiments, we use the two moons dataset with three types of input shift: shear, rotation, and translation (details in Appendix D.4). For real data with synthetic corruptions, we use CIFAR-10C [Hendrycks and Dietterich, 2019], which applies algorithmic covariate corruptions to the CIFAR-10 test set across five severity levels (1–5). To evaluate under a real distribution shift, we also consider Camelyon17-WILDS [Koh et al., 2021]—a cancer detection dataset where test data is collected from a different hospital system than the training data.

Table 1: **Experiments on calibration across model classes on CIFAR-100**. Temperature scaling (TEMP) significantly improves ECE over the Maximum Softmax Probability (MSP) baseline but does not help to close the selective classification gap. Self-Adaptive Training (SAT) and Deep Ensembles (DE) improve calibration non-monotonically and also improve selective classification acceptance ordering through re-ranking. A corresponding plot is given in Figure 5; more datasets in Tables 2, 3.

| | CNN | | | | ResNet-18 | | | | WideResNet-50 | | | |
| --- | --- | --- | --- | --- | --- | --- | --- | --- | --- | --- | --- | --- |
| | MSP | TEMP | SAT | DE | MSP | TEMP | SAT | DE | MSP | TEMP | SAT | DE |
| E-AURC | 0.086 | 0.085 | 0.081 | 0.065 | 0.033 | 0.033 | 0.028 | 0.026 | 0.031 | 0.032 | 0.028 | 0.026 |
| ECE | 0.142 | 0.008 | 0.116 | 0.019 | 0.052 | 0.048 | 0.026 | 0.034 | 0.066 | 0.050 | 0.046 | 0.030 |

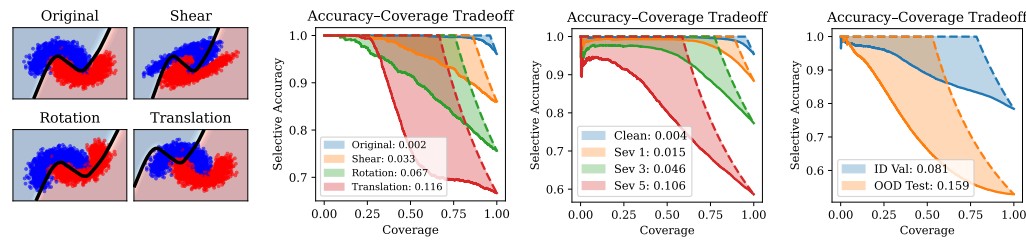

(a) Distribution shifts with two moons dataset.      (b) CIFAR-10C      (c) Camelyon17-WILDS

Figure 4: **Experiments on distribution shifts**. We find that shifts can also significantly contribute to the gap. (a) Two moons under shear, rotation, and translation with corresponding accuracy–coverage curves. (b) CIFAR-10C across three distinct corruption severities. (c) Camelyon17 OOD shift.

*Findings.* Figure 4 shows a clear trend: as covariate shifts intensify, the accuracy–coverage curve moves farther below its oracle bound, indicating that abstention no longer isolates easy inputs. Selective classifiers thus become *over-confidently wrong*, echoing evidence that many uncertainty metrics deteriorate under shift or misspecification [Snoek et al., 2019]. As the gap grows with shift severity, deployments must pair selective prediction with robust ranking or shift-detection safeguards.

## 5 Conclusion

Building a truly performant selective classifier hinges on understanding and closing the gap between practical models and the oracle perfect-ordering bound. To answer *what it takes*, we introduce a coverage-uniform selective-classification gap and derive the first finite-sample decomposition that pinpoints exactly five limiting factors: three intrinsic sources—Bayes noise, approximation error, and ranking (calibration) error—and two contingent slack terms—sampling variability and implementation or distribution-shift artifacts. Our experiments show that each component can be individually measured and, importantly, directly improved: stronger model backbones reduce approximation error, non-monotone or feature-aware scoring shrinks ranking error, and shift-robust training with larger validation sets minimizes residual slack. Together, these insights provide a clear recipe for designing and evaluating high-performance selective classifiers.

**Limitations and future work.** While our decomposition cleanly bounds the selective-classification gap, its error budgets can *interact*—for example, increasing capacity often improves both approximation and ranking—which makes unique attribution challenging. Many *training-time calibration schemes* (e.g., SAT, mixup, focal loss) simultaneously affect ranking and full-coverage accuracy, confounding the separation of budgets. Our core experiments focus on *synthetic and vision benchmarks*; extending these insights to large-scale foundation models would be an important direction. We present a preliminary exploration on large language models in Appendix F.2. Finally, because our oracle bound and gap are defined for *0–1 loss*, adapting to *asymmetric or class-dependent cost functions*—often required in high-stakes decision-making—will require generalizing both the bound and its decomposition. Our finite-sample gap decomposition lays the groundwork for a more unified reliability framework; extending it to (i) settings where out-of-distribution inputs must be rejected and (ii) open-ended language generation constitutes a promising agenda for future work.

## Acknowledgements

We acknowledge the following sponsors, who support our research with financial and in-kind contributions: Apple, CIFAR through the Canada CIFAR AI Chair, Meta, NSERC through the Discovery Grant and an Alliance Grant with ServiceNow and DRDC, the Ontario Early Researcher Award, the Schmidt Sciences foundation through the AI2050 Early Career Fellow program. Resources used in preparing this research were provided, in part, by the Province of Ontario, the Government of Canada through CIFAR, and companies sponsoring the Vector Institute. We thank Relu Patrascu and the computing team at the University of Toronto's Computer Science Department for administrating and procuring the compute infrastructure used for the experiments in this paper. We would also like to thank Andy Wei Liu, Anvith Thudi, David Glukhov, Vardan Papyan, and many others at the Vector Institute for discussions contributing to this paper.

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

## A  Broader Impact

This work introduces a decomposition of the selective-classification gap into measurable components—Bayes noise, approximation error, ranking error, statistical noise, and deployment slack—offering practical guidance for improving abstaining classifiers. By diagnosing which source dominates in a given setting, our method supports more targeted model design and evaluation.

**Positive implications.**  Our decomposition improves transparency and supports safer deployment in high-stakes domains by helping practitioners understand whether their model underperforms due to ranking, capacity, or robustness. Because each gap component is explicitly quantified, our approach can serve as a tool for model debugging, monitoring, and fairer benchmarking.

**Potential risks.**  Selective classifiers may disproportionately defer on certain groups, amplifying disparities—a risk previously observed by Jones et al. [2021]. Additionally, institutions may exploit uncertainty estimates to justify *strategic abstention*—deliberately deferring on individuals they prefer not to serve [Rabanser et al., 2025]. While our framework identifies which part of the gap drives poor performance, it does not control how deferred inputs are handled.

**Mitigations.**  We recommend reporting gap components disaggregated by sensitive attributes, auditing scoring functions for spurious correlations, and documenting fallback policies. These steps are essential to ensure that abstention mechanisms improve reliability without undermining fairness.

**Outlook.**  We hope this work encourages more precise evaluations of selective classifiers, shifting focus from aggregate calibration to interpretable, component-wise gap analysis that can inform both technical improvements and policy safeguards.

## B  Methods Extension

### B.1  Detailed Proof of Theorem 1

We restate the theorem for convenience.

---

**Theorem 2** (Selective classification Gap; detailed). Fix a coverage level $c \in (0,1]$, a score function $g(\cdot, h)$, and its associated population threshold $t_c$ satisfying $\Pr\big(g(X, h) \geq t_c\big) = c$. Define the accepted region $A_c := \{x : g(x, h) \geq t_c\}$ and the oracle region $A_c^\star := \{x : \eta_h(x)$ is among the largest $c$-fraction$\}$. With the error terms

$$\varepsilon_{\text{Bayes}}(c) = \mathbb{E}\left[1 - \max\{\eta(X), 1 - \eta(X)\} \mid X \in A_c\right], \tag{21}$$

$$\varepsilon_{\text{approx}}(c) = \mathbb{E}\left[|\eta_h(X) - \eta(X)| \mid X \in A_c\right], \tag{22}$$

$$\varepsilon_{\text{rank}}(c) = \mathbb{E}\left[\eta_h(X) \mid X \in A_c^\star\right] - \mathbb{E}\left[\eta_h(X) \mid X \in A_c\right] \ (\geq 0), \tag{23}$$

the population gap satisfies

$$\Delta(c) = \overline{\text{acc}}\big(a_{\text{full}}, c\big) - \text{acc}_c(h, g) \ \leq \ \varepsilon_{\text{Bayes}}(c) + \varepsilon_{\text{approx}}(c) + \varepsilon_{\text{rank}}(c). \tag{24}$$

Moreover, let $\widehat{\Delta}(c)$ be the empirical gap computed on $n$ independent test samples. Then for any $\delta \in (0,1)$, with probability at least $1 - \delta$,

$$\widehat{\Delta}(c) \ \leq \ \varepsilon_{\text{Bayes}}(c) + \varepsilon_{\text{approx}}(c) + \varepsilon_{\text{rank}}(c) + C\sqrt{\frac{\log(1/\delta)}{n}}, \tag{25}$$

where $C > 0$ is an absolute constant.

**Proof.** We split the argument into four self-contained steps.

**Step 0.  Oracle upper bound revisited.** For completeness we justify the piecewise form of $\overline{\text{acc}}\big(a_{\text{full}}, c\big)$ in Definition 3. Because $a_{\text{full}} = \Pr(h(X) = Y) = \mathbb{E}[\eta_h(X)]$, the set $\{x : \eta_h(x) = 1\}$ has probability mass at least $a_{\text{full}}$. Hence an oracle that retains the

---

highest-confidence points achieves perfect accuracy for all coverages $c \le a_{\text{full}}$. For $c > a_{\text{full}}$, the best it can do is include *only* those perfect points plus a $(c - a_{\text{full}})$-fraction of the remaining examples, which contribute at worst zero accuracy. Therefore

$$\overline{\text{acc}}(a_{\text{full}}, c) = \frac{a_{\text{full}}}{c}, \qquad a_{\text{full}} < c < 1. \tag{26}$$

**Step 1. Algebraic decomposition of the gap.** Recall that $\text{acc}_c(h, g) = \mathbb{E}[\eta_h(X) \mid X \in A_c]$. We repeatedly add and subtract the same quantity:

$$
\begin{aligned}
\Delta(c) &:= \overline{\text{acc}}(a_{\text{full}}, c) - \text{acc}_c(h, g) \\
&= \overline{\text{acc}}(a_{\text{full}}, c) - \mathbb{E}[\eta_h \mid A_c^\star] + \mathbb{E}[\eta_h \mid A_c^\star] - \mathbb{E}[\eta_h \mid A_c] \\
&\le \mathbb{E}[\eta_h \mid A_c^\star] - \mathbb{E}[\eta_h \mid A_c] && \text{(rank)} \\
&\quad + \mathbb{E}[\eta_h - \mathbb{I}_{\{h=Y\}} \mid A_c] && \text{(approx+Bayes)} \\
&= \varepsilon_{\text{rank}}(c) + \varepsilon_{\text{approx}}(c) + \varepsilon_{\text{Bayes}}(c). && (27)
\end{aligned}
$$

**Explanation of the two labelled inequalities.**

1. **(rank)** isolates the ranking error, $\varepsilon_{\text{rank}}(c) := \mathbb{E}[\eta_h \mid A_c^\star] - \mathbb{E}[\eta_h \mid A_c]$. The inequality holds because the remaining term from the previous line, $\overline{\text{acc}}(a_{\text{full}}, c) - \mathbb{E}[\eta_h \mid A_c^\star]$, is a non-negative quantity that is bounded by the error sources introduced next.

2. **(approx+Bayes)** adds and subtracts $\eta(X)$ inside the expectation, then splits the absolute value:
$$\eta_h - I_{\{h=Y\}} = (\eta_h - \eta) + (\eta - I_{\{h=Y\}}). \tag{28}$$
The second summand satisfies the deterministic bound $|\eta(X) - I_{\{h=Y\}}| = \max\{\eta, 1 - \eta\} - I_{\{h=Y\}} \le 1 - \max\{\eta, 1 - \eta\}$, yielding exactly $\varepsilon_{\text{Bayes}}(c)$. The first summand contributes $\varepsilon_{\text{approx}}(c)$.

**Step 2. Non-negativity of $\varepsilon_{\text{rank}}(c)$.** Because $\eta_h(X) \in [0, 1]$ and $A_c^\star$ contains the $c$-fraction of points with the largest $\eta_h$-values, $\mathbb{E}[\eta_h \mid A_c^\star] \ge \mathbb{E}[\eta_h \mid A_c]$, hence $\varepsilon_{\text{rank}}(c) \ge 0$ as stated.

**Step 3. Finite-sample deviation.** Let $\widehat{\mu}$ be any empirical average of a $[0, 1]$-valued random variable with expectation $\mu$. Hoeffding's inequality gives $\Pr(|\widehat{\mu} - \mu| > \epsilon) \le 2e^{-2n\epsilon^2}$. Apply this bound separately to the three empirical estimates that constitute $\widehat{\Delta}(c)$, and take a union bound with $\epsilon = \sqrt{\frac{\log(6/\delta)}{2n}}$. This yields, with probability at least $1 - \delta$, $|\widehat{\Delta}(c) - \Delta(c)| \le C\sqrt{\log(1/\delta)/n}$ for an absolute constant $C$. Combining with (24) proves (25).

**Step 4. Connection to ranking distance.** Define the mass of mis-ordered points $D_{\text{rank}}(c) := \Pr(X \in A_c^\star \setminus A_c) + \Pr(X \in A_c \setminus A_c^\star)$. Because $\eta_h \in [0, 1]$,

$$\varepsilon_{\text{rank}}(c) = \mathbb{E}[\eta_h \mid A_c^\star] - \mathbb{E}[\eta_h \mid A_c] \tag{29}$$

$$\le \|\eta_h\|_\infty D_{\text{rank}}(c) \tag{30}$$

$$\le D_{\text{rank}}(c). \tag{31}$$

Hence $\varepsilon_{\text{rank}}(c) = 0$ if and only if $A_c = A_c^\star$.

This completes the proof. $\qquad\qquad\square$

**Multiclass remark.** For $K > 2$ labels, define $\eta(x) = \big(\Pr(Y = 1 \mid x), \ldots, \Pr(Y = K \mid x)\big)$ and its complement confidence $\eta^{\max}(x) = \max_k \eta_k(x)$. Then the inequality $|\eta^{\max} - I_{\{h=Y\}}| \le 1 - \eta^{\max}$ replaces the binary bound above, and the rest of the argument goes through verbatim. The approximation term becomes $\mathbb{E}[\|\eta_h - \eta\|_1 \mid A_c]$; all other quantities are unchanged.

## B.2 When Can Temperature Scaling Re-rank Confidence Scores?

Temperature scaling multiplies every logit by the same factor $1/T$ $(T > 0)$ before the softmax,

$$p_j^{(T)}(x) = \frac{\exp(z_j(x)/T)}{\sum_k \exp(z_k(x)/T)}. \tag{32}$$

Although the predicted label $\arg\max_j z_j(x)$ is invariant to $T$, the *confidence score* $s_T(x) = \max_j p_j^{(T)}(x)$ can change its *cross-sample* ordering.

**General form.** Let $j_\star = \arg\max_j z_j(x)$ and $r_j(x) = \exp\big(z_j(x) - z_{j_\star}(x)\big)$ $(j \neq j_\star)$. Then

$$s_T(x) = \frac{1}{1 + \sum_{j \neq j_\star} r_j(x)^{1/T}}. \tag{33}$$

For binary classification, the sum has a single term and (33) collapses to the familiar logistic form $s_T(x) = 1/(1 + e^{-\Delta/T})$ with $\Delta = z_{j_\star} - z_{3-j_\star}$.

**Two-sample condition.** For two inputs $x_1, x_2$ let $S_i(T) = \sum_{j \neq j_\star^{(i)}} r_{ij}^{1/T}$. Because each $r_{ij} \leq 1$, every $r_{ij}^{1/T}$ is monotone non-decreasing in $T$ (strictly increasing unless there is a tie), and the ordering $s_T(x_1) > s_T(x_2)$ can change exactly at those temperatures $T$ where $S_1(T) = S_2(T)$.

**Illustrative example ($K = 3$).**

$$z^{(1)} = (-2, -3, -3), \quad z^{(2)} = (0, -0.1, -3). \tag{34}$$

At $T = 1$ one finds $s_1(x_1) = 0.576 > 0.512 = s_1(x_2)$, while at $T = 3$ we see that $s_3(x_1) = 0.411 < 0.428 = s_3(x_2)$, so temperature scaling would now accept $x_2$ before $x_1$.

**How likely is a swap?** Equation (33) shows that a swap requires the one-dimensional curves $S_1(T)$ and $S_2(T)$ to intersect. Since the curves are continuous and monotone, the intersection occurs— if at all—at isolated temperatures and only when the competing logit patterns are finely tuned.

**Practical implication.** Temperature scaling can *in principle* tighten the selective-classification gap, but only for the vanishingly small subset of inputs whose non-maximum logits happen to satisfy $S_1(T^\star) = S_2(T^\star)$. To obtain a meaningful re-ordering one must therefore adopt *non-monotone* calibration strategies.

## B.3 Additional Contingent Slack

In the main text (Sec. 3.5) we folded all implementation-level imperfections into a single residual term $\varepsilon_{\text{misc}}(c)$, retaining only optimization error and distribution shift explicitly. Here we list two further slack terms omitted there:

3. **Threshold-selection noise $\varepsilon_{\text{thr}}(c)$.**
   When the coverage threshold $\hat{t}_c$ is chosen on a validation set of size $m$, the realized coverage deviates from the target $c$ by
   $$O\big(\sqrt{c(1-c)/m}\big), \tag{35}$$
   inducing a corresponding vertical shift in selective accuracy.

4. **Tie-breaking / score quantization $\varepsilon_{\text{tie}}(c)$.**
   Discrete confidence values (e.g. low-precision logits) create equivalence classes of samples with identical scores. If $\kappa$ denotes the maximum number of tied samples at any score level, then
   $$\varepsilon_{\text{tie}}(c) \leq \frac{\kappa}{n}, \tag{36}$$
   where $n$ is the size of the evaluation set.

**Residual slack revisited.** Together with optimization error $\varepsilon_{\text{opt}}$ and shift $\varepsilon_{\text{shift}}(c)$, these yield

$$\varepsilon_{\text{misc}}(c) = \varepsilon_{\text{opt}} + \varepsilon_{\text{shift}}(c) + \varepsilon_{\text{thr}}(c) + \varepsilon_{\text{tie}}(c). \tag{37}$$

# C Practitioner Checklist for Tightening the Selective-Classification Gap

Below is an expanded, actionable checklist to help practitioners systematically tackle each component of the selective-classification gap. For each item, we list concrete steps, recommended tools, and pointers to reduce the corresponding error term.

- $\varepsilon_{\mathbf{approx}}$ **— Shrink Approximation Error**

  - *Model capacity:* Upgrade to deeper or wider architectures like ResNeXt, ViT, or ConvNeXt to better approximate complex functions and reduce base error [Xie et al., 2017, Dosovitskiy et al., 2021, Liu et al., 2022, Kadavath et al., 2022].

  - *Pre-training:* Initializing with rich features from self-supervised methods (SimCLR, BYOL) or foundation models (CLIP, DINO) can improve out-of-the-box performance, convergence, and uncertainty scores [Chen et al., 2020, Grill et al., 2020, Radford et al., 2021, Caron et al., 2021, Hendrycks et al., 2019]. However, pre-training can also sometimes negatively affect selective classification performance [Galil et al., 2023].

  - *Distillation:* Use teacher–student training with logit matching or feature hints to inherit accuracy from a larger model at lower cost [Galil et al., 2023, Hinton et al., 2015, Dietmüller et al., 2024].

  - *Data augmentation:* Augmentations can often improve generalization with policy-based (AutoAugment, RandAugment) or mixing-based (MixUp, CutMix) augmentations to regularize the learner [Cubuk et al., 2018, 2020, Zhang et al., 2018, Yun et al., 2019]. However, strong augmentations may also degrade selective classification performance for certain minority classes [Jones et al., 2021].

- $\varepsilon_{\mathbf{rank}}$ **— Improve Ranking Calibration:**

  - *Feature-aware scoring:* Train auxiliary heads like ConfidNet to learn correctness scores using both logits and input features [Corbière et al., 2019], often improving uncertainty estimates. Self-Adaptive Training (SAT) further enhances this by encouraging internal representations to separate correct and incorrect predictions through contrastive regularization or supervised signals [Huang et al., 2020].

  - *Deep ensembles:* Use the disagreement or predictive entropy across multiple independently trained models to estimate uncertainty [Lakshminarayanan et al., 2017].

  - *Conformal methods:* Generate conformal p-values or risk-controlled selection sets that respect desired coverage guarantees [Vovk et al., 2005, Angelopoulos et al., 2024].

  - *Use caution with vector/Dirichlet scaling:* While previous work has shown that vector, matrix, or Dirichlet transformations can be beneficial to reshape confidence distributions [Guo et al., 2017, Kull et al., 2019], Le-Coz et al. [2024] shows that these techniques can harm ranking under a large number of classes.

- $\varepsilon_{\mathbf{opt}}$ **— Reduce Optimization Error:**

  - *Convergence diagnostics:* Track training/validation loss curves to detect underfitting and determine optimal stopping points [Salakhutdinov, 2014].

  - *Learning-rate schedules:* Employ dynamic LR strategies like cosine decay, OneCycle, or CLR to reach better optima more consistently [Loshchilov and Hutter, 2017, Smith and Topin, 2019, Smith, 2017].

  - *Early stopping / checkpoints:* Save and average late-stage checkpoints or use snapshot ensembling to smooth optimization variance [Huang et al., 2017, Lakshminarayanan et al., 2017, Rabanser et al., 2022].

  - *Regularization:* Use dropout, weight decay, or stochastic depth to prevent overfitting and stabilize training [Srivastava et al., 2014, Huang et al., 2016, Loshchilov et al., 2017].

- $\varepsilon_{\mathbf{Bayes}}$ **— Quantify Irreducible Noise:**

  - *Repeated labels:* Collect multiple annotations (e.g., CIFAR-10H) to estimate human-level disagreement and the Bayes error floor [Peterson et al., 2019, Wei et al., 2022].

  - *Noise-robust training:* Mitigate label noise using bootstrapped or Taylor-truncated loss functions that temper reliance on hard labels [Reed et al., 2014, Feng et al., 2020].

  - *Dataset curation:* Apply confident learning to flag likely label errors or use active learning for data relabeling [Northcutt et al., 2021].

- $\varepsilon_{\mathbf{stat}}$ **— Control Statistical Slack:**
  - *Validation set size:* Use a sufficiently large holdout set to estimate thresholds and calibrate uncertainty reliably [Hart et al., 2001].
  - *Confidence intervals:* Use DKW or Clopper–Pearson bounds to set conservative thresholds with statistical guarantees on coverage [Massart, 1990, Clopper and Pearson, 1934].
  - *Cross-validation:* Average selection thresholds over folds to reduce their variance and avoid overfitting to a single validation set [Kohavi et al., 1995].

- $\varepsilon_{\mathbf{shift}}$ **— Mitigate Distribution Shift:**
  - *Shift detection:* Detect covariate shift via statistical two-sample tests such as MMD or KL divergence between feature distributions [Gretton et al., 2012, Rabanser et al., 2019].
  - *Importance weighting:* Correct mismatched data distributions with density ratio weighting, e.g., using kernel mean matching [Huang et al., 2006].
  - *Domain adaptation:* Finetune with in-domain examples or use unsupervised techniques like AdaBN or domain-adversarial training (DANN) [Ganin et al., 2016, Li et al., 2016].
  - *Test-time adaptation:* Adapt models at inference using entropy minimization (Tent) or batch norm recalibration to restore accuracy under shift [Nado et al., 2021, Wang et al., 2021].

- $\varepsilon_{\mathbf{thr}}$ **— Threshold–Selection Noise:**
  - *Bootstrap resampling:* Estimate variability in the selection threshold $\tau_c$ by computing its standard error across bootstrap samples [Tibshirani and Efron, 1993].
  - *Smooth thresholds:* Interpolate between adjacent scores or accept a random subset at the threshold to reduce coverage discontinuities [Angelopoulos and Bates, 2021].

- $\varepsilon_{\mathbf{tie}}$ **— Tie-Breaking & Score Quantization:**
  - *Higher precision:* Use higher float precision (e.g., FP32 or FP64) or more logits bits to distinguish close scores and avoid ties [Micikevicius et al., 2018].
  - *Dithering:* Add tiny random noise to scores before thresholding to stochastically resolve ties and reduce instability.
  - *Refrain from binning:* Histogram binning (HQ) or Bayesian Binning into Quantiles (BBQ) often improve calibration but not selective classification performance [Naeini et al., 2015, Le-Coz et al., 2024].

**Putting it all together.** After addressing each bullet above, recompute your selective accuracy–coverage curve and compare to the oracle bound (Def. 3). Iterating over these steps will systematically shrink $\widehat{\Delta}(c)$ toward its irreducible floor.

# D   Experimental Details

## D.1   Computational Resources

Our experiments were conducted on a mix of GPU-equipped compute nodes with varying hardware configurations. Some machines are equipped with Intel Xeon Silver CPUs (10 cores, 20 threads) and 128GB of RAM, each hosting 4× NVIDIA GeForce RTX 2080 Ti GPUs with 11GB VRAM. Others feature AMD EPYC 7643 processors (48 cores, 96 threads), 512GB of RAM, and 4× NVIDIA A100 GPUs, each with 80GB VRAM.

## D.2   Hyper-Parameters

We follow standard literature-recommended training settings across all datasets. For each architecture–dataset pair, we use a fixed learning rate, weight decay, and batch size as detailed below:

- **SimpleCNN:**
  - Learning rate: 0.01
  - Weight decay: $1 \times 10^{-4}$
  - Batch size: 128
- **ResNet-18:**

- Learning rate: 0.1 for CIFAR datasets; 0.01 for Stanford Cars, Camelyon17
- Weight decay: $5 \times 10^{-4}$
- Batch size: 128

- **WideResNet-50-2:**
  - Same settings as ResNet-18

- **Epochs:**
  - 200 epochs for all datasets except Camelyon17, which uses 10

- **Optimization:** SGD with momentum 0.9, Nesterov enabled, and a cosine annealing learning rate schedule.

- **Selective prediction methods:**
  - `MSP`: Standard cross-entropy training
  - `SAT`: Cross-entropy pretraining for half of training epochs, followed by Self-Adaptive Training (momentum 0.9) with an extra abstain class

All experiments use fixed random seeds for reproducibility and standard data augmentation per dataset (random crops, flips, normalization).

### D.3 SimpleCNN Architecture

The SimpleCNN model is a compact convolutional neural network used for experiments on lower-resolution image datasets. The architecture is defined by the following sequence of layers:

- A $3 \times 3$ convolution with 32 filters and padding 1, followed by ReLU and $2 \times 2$ max-pooling.
- A second $3 \times 3$ convolution with 64 filters and padding 1, followed by ReLU and $2 \times 2$ max-pooling.
- A flattening layer, followed by a fully connected layer with 128 hidden units and ReLU activation.
- A final fully connected layer projecting to the number of output classes.

Let $s = \texttt{input\_size}//4$ denote the spatial resolution after two $2 \times 2$ pooling layers. Then, the full model is:

$$
\begin{aligned}
\texttt{SimpleCNN}(x) = {} & \texttt{Linear}\big(128 \to \texttt{num\_classes}\big) \circ \texttt{ReLU} \circ \\
& \texttt{Linear}\big(64 \cdot s^2 \to 128\big) \circ \texttt{Flatten} \circ \\
& \texttt{MaxPool2d} \circ \texttt{ReLU} \circ \texttt{Conv2d}(32 \to 64) \circ \\
& \texttt{MaxPool2d} \circ \texttt{ReLU} \circ \texttt{Conv2d}(3 \to 32)(x)
\end{aligned}
$$

The number of output classes is set as follows:

$$
\texttt{num\_classes} = \begin{cases} 10 & \text{for CIFAR-10,} \\ 100 & \text{for CIFAR-100,} \\ 196 & \text{for Stanford Cars,} \\ 2 & \text{for Camelyon17,} \end{cases} \quad \text{with an optional extra class if } \texttt{extra\_class} \text{ is True.}
$$

The input size is dataset-dependent and set to:

$$
\texttt{input\_size} = \begin{cases} 32 & \text{for CIFAR-10 and CIFAR-100,} \\ 224 & \text{for Stanford Cars, Camelyon17.} \end{cases}
$$

The model structure is summarized below:

```
SimpleCNN(
  (net): Sequential(
    (0): Conv2d(3, 32, kernel_size=(3, 3), stride=(1, 1), padding=1)
    (1): ReLU()
```

```
5        (2): MaxPool2d(kernel_size=2, stride=2)
6        (3): Conv2d(32, 64, kernel_size=(3, 3), stride=(1, 1), padding=1)
7        (4): ReLU()
8        (5): MaxPool2d(kernel_size=2, stride=2)
9        (6): Flatten(start_dim=1)
10       (7): Linear(in_features=4096, out_features=128)
11       (8): ReLU()
12       (9): Linear(in_features=128, out_features=num_classes)
13     )
14   )
```

### D.4 Synthetic Distribution Shifts on Two Moons

To evaluate robustness under controlled covariate shifts, we apply a series of synthetic affine transformations to the test set of the standard two moons dataset. Each transformation simulates a distinct type of distribution shift:

- **Original:** No transformation; the unperturbed test set.
- **Shear:** A shear transformation along the $x$-axis defined by:

$$\text{Shear matrix} \quad S = \begin{bmatrix} 1 & 1.25 \\ 0 & 1 \end{bmatrix}, \quad \text{so that} \quad x' = Sx = \begin{bmatrix} x + 1.25y \\ y \end{bmatrix}. \quad (38)$$

- **Rotation:** A rotation by 30 degrees counterclockwise, using:

$$R = \begin{bmatrix} \cos\theta & -\sin\theta \\ \sin\theta & \cos\theta \end{bmatrix}, \quad \theta = \frac{\pi}{6}. \quad (39)$$

- **Translation:** A shift of the input space by a fixed vector:

$$x' = x + t, \quad \text{where} \quad t = \begin{bmatrix} 1.0 \\ -0.5 \end{bmatrix}. \quad (40)$$

Each transformation is applied to the test data matrix $X_{\text{test}}$ via matrix multiplication or translation, yielding the following test sets:

$$\begin{aligned} \text{Original:} &\quad X_{\text{test}} \\ \text{Shear:} &\quad X_{\text{test}} \cdot S^{\top} \\ \text{Rotation:} &\quad X_{\text{test}} \cdot R^{\top} \\ \text{Translation:} &\quad X_{\text{test}} + t \end{aligned} \quad (41)$$

These transformations create meaningful distribution shifts while preserving label semantics, enabling precise evaluations of model robustness under shift.

### D.5 CIFAR-10C Severity Levels

For the CIFAR-10C severity levels (1–5), we aggregate all 15 corruption types at a given severity to form a single validation set. For severity level $l$, we collect all corruptions labeled as severity $l$ across the following categories:

- **Noise:** `gaussian_noise`, `shot_noise`, `impulse_noise`
- **Blur:** `defocus_blur`, `glass_blur`, `motion_blur`, `zoom_blur`
- **Weather:** `snow`, `frost`, `fog`, `brightness`
- **Digital:** `contrast`, `elastic_transform`, `pixelate`, `jpeg_compression`

This results in a single validation set per severity level $l$, where each image is sampled from one of these 15 corruptions applied at the specified severity.

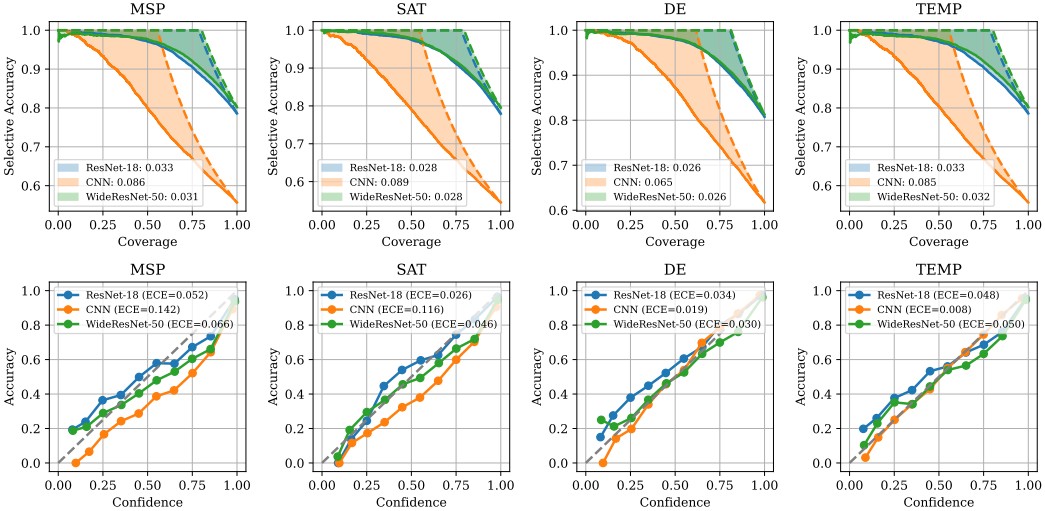

Figure 5: **Comparison between gap and calibration on CIFAR-100.** *Top*: selective accuracy curves across four training methods and three architectures. *Bottom*: corresponding reliability diagrams (ECE shown in parentheses). Temperature scaling (TEMP) consistently improves calibration but does not reduce the gap. By contrast, SAT and DE reduce the gap more effectively—especially for larger models—by improving the ranking.

## E   Loss Prediction, Multicalibration, and Ranking Error

This appendix offers an alternative perspective on the ranking error term $\varepsilon_{\text{rank}}(c)$ by framing it as a challenge of per-example loss prediction. Instead of building directly on the calibration discussion in Section 3.4, we show how the ability to forecast one's own 0–1 loss tightly controls the selective-classification gap. We formalize this connection through the recent theory of loss prediction [Gollakota et al., 2025] and multicalibration [Hébert-Johnson et al., 2018]. Throughout we adopt the binary-label conventions of Section 3.3. Extensions to multiclass losses likewise follow by one-vs-rest reduction.

### E.1   Loss-Prediction Preliminaries

Let $\ell(h(x), y) = \mathbb{I}\{h(x) \neq y\}$ denote the 0-1 loss of a fixed classifier $h$. A *loss predictor* $\text{LP} \colon \Phi \to \mathbb{R}$ maps auxiliary features $\phi(x, h) \in \Phi$ to an estimate of $\ell(h(x), y)$. The canonical baseline is the *self-entropy predictor* $\text{SEP}(x) := \mathbb{E}[\ell(h(x), y) \mid h(x)]$ (which equals $\min\{p, 1-p\}$ for probabilistic $p = h(x)$).

> **Definition 6** (Advantage over the self-entropy predictor). The (squared-error) *advantage* of a loss predictor LP is
>
> $$\text{Adv}(\text{LP}) := \mathbb{E}\big[(\ell - \text{SEP})^2\big] - \mathbb{E}\big[(\ell - \text{LP})^2\big]. \tag{42}$$
>
> A positive advantage means LP forecasts the instance-wise loss better than the model itself.

Depending on $\phi$, we obtain a hierarchy of predictors: prediction-only ($\phi = h(x)$), input-aware ($\phi = (h(x), x)$), and representation-aware ($\phi = (h(x), x, r(x))$); we refer to Gollakota et al. [2025] for a detailed taxonomy.

### E.2   Multicalibration Background

Multicalibration is a fine-grained notion of reliability that asks not just for global calibration, but for calibration conditional on a rich class of subpopulations or features [Hébert-Johnson et al., 2018]. At a high level, a model is multicalibrated if its predicted scores match outcomes not only on average, but also across a large collection of subsets defined by auxiliary variables or internal representations.

> **Definition 7** (Multicalibration Error). Let $C$ be a class of weighting functions $c \colon \Phi \to [-1, 1]$, and let $h \colon \mathcal{X} \to [0, 1]$ be a classifier. The *multicalibration error* of $h$ with respect to $C$ is defined as
> $$\mathrm{MCE}(C, h) \; := \; \max_{c \in C} \Big| \mathbb{E}\big[ (Y - h(X))\, c(\phi(X, h)) \big] \Big|. \tag{43}$$

Each function $c \in C$ defines a subpopulation or slice of the input space via its support. The quantity $\mathrm{MCE}(C, h)$ measures how well the model's predicted scores $h(x)$ match the true label $Y$ when weighted over these slices. When $C$ consists of indicator functions over discrete demographic subgroups, small $\mathrm{MCE}(C, h)$ implies groupwise calibration. More generally, if $C$ includes continuous or data-dependent functions (e.g., based on internal features), low multicalibration error guarantees alignment between predicted and true outcomes across a flexible set of conditions.

In our selective classification setting, $\phi(x, h)$ may include the model's output confidence, the input $x$, or hidden representations from the network. The class $C$ can be constructed accordingly to enforce calibration in feature-dependent or risk-sensitive regions of the input space.

### E.3  Loss Prediction $\iff$ Multicalibration

We now describe how the ability to predict one's own 0–1 loss is deeply connected to multicalibration. This perspective stems from the work of Gollakota et al. [2025], who characterize when a model "knows its own loss" in terms of multicalibration violations.

Let $F$ be a class of loss predictors $\mathrm{LP} \colon \phi(x, h) \mapsto \hat{\ell} \in [0, 1]$, which estimate the 0–1 loss $\ell(h(x), y) = \mathbb{I}\{h(x) \neq y\}$ of a fixed classifier $h$. As discussed in Section E.1, a loss predictor is considered good if it has a significant squared-error advantage over the model's self-estimate $\mathrm{SEP}(x)$.

Remarkably, Gollakota et al. [2025] show that this predictive advantage is tightly characterized by the multicalibration error of the model—measured over a derived weight class $C$ that depends on the predictors in $F$. The following theorem formalizes this connection:

> **Theorem 3** (Gollakota et al. [2025], Thm. 4.1—adapted). For any function class $F$ of loss predictors and the associated weight class $C = \{(f - \mathrm{SEP}) \cdot H'_\ell(h(x)) : f \in F\}$,
> $$\tfrac{1}{2} \max_{\mathrm{LP} \in F} \mathrm{Adv}(\mathrm{LP}) \;\leq\; \mathrm{MCE}(C, h) \;\leq\; \sqrt{\max_{\mathrm{LP} \in F'} \mathrm{Adv}(\mathrm{LP})}, \tag{44}$$
> where $F'$ augments $F$ with linear mixtures of SEP and elements of $F$. Thus a non-trivial advantage is possible *iff* $h$ exhibits a multicalibration violation of similar magnitude.

This result bridges two domains: learning to predict loss (a regression task) and satisfying a generalization constraint (calibration under distributional conditions). In the selective classification setting, this insight underpins Corollary 1, which shows that the ranking error—and hence the gap to oracle performance—is tightly controlled by the model's ability to forecast its own mistakes.

### E.4  Bounding the Ranking-Error Term $\varepsilon_{\mathbf{rank}}(c)$

Theorem 3 translates into a bound on the ranking error that drives the selective-classification gap.

> **Corollary 1** (Loss-prediction advantage controls mis-ranking). Fix coverage $c \in (0, 1]$ and let $\mathrm{Adv}^\star := \max_{\mathrm{LP} \in F} \mathrm{Adv}(\mathrm{LP})$ for some input-aware class $F$. Then the ranking-error term in Theorem 1 satisfies $\varepsilon_{\mathrm{rank}}(c) \;\leq\; \sqrt{2\, \mathrm{Adv}^\star}$.

> **Proof.** Recall that $A_c^\star = \{x : \eta_h(x) \text{ is in the top } c\text{-mass}\}$ and $A_c = \{x : g(x, h) \geq t_c\}$. Write the *difference indicator* $\delta_c(x) := \mathbb{I}_{A_c^\star}(x) - \mathbb{I}_{A_c}(x) \in \{-1, 0, 1\}$ so $\Pr(\delta_c = 1) = \Pr(\delta_c = -1) = c$ and $\mathbb{E}[\delta_c] = 0$.

**Step 1: Express ranking error as a covariance.** With $r(x) := \mathbb{I}\{h(x) = Y\}$ we have

$$\varepsilon_{\text{rank}}(c) = \mathbb{E}[r \mid A_c^\star] - \mathbb{E}[r \mid A_c] = \frac{1}{c}\,\mathbb{E}\big[r(X)\,\delta_c(X)\big]. \qquad (45)$$

**Step 2: Replace correctness by residual** $Y - h(X)$**.** Because $r = 1 - \ell$ and $\ell = (Y - h)^2$ for binary labels,

$$r\,\delta_c = \big(1 - (Y - h)^2\big)\delta_c = -(Y - h)\,\delta_c \quad (\text{since } \mathbb{E}[\delta_c] = 0). \qquad (46)$$

Hence

$$\varepsilon_{\text{rank}}(c) = \frac{1}{c}\,\big|\mathbb{E}[(Y - h(X))\,\delta_c(X)]\big|. \qquad (47)$$

**Step 3: Bound the covariance by multicalibration error.** Define the bounded weight function $c^\star(x) := \delta_c(x)$; then $|c^\star(x)| \leq 1$, so $c^\star \in C$ (the weight class in Theorem 3). By definition of multicalibration error,

$$\big|\mathbb{E}[(Y - h(X))\,c^\star(X)]\big| \;\leq\; \text{MCE}(C, h). \qquad (48)$$

Combining (47) and (48) with $c \leq 1$ yields

$$\varepsilon_{\text{rank}}(c) \;\leq\; \text{MCE}(C, h). \qquad (49)$$

**Step 4: Invoke the loss-prediction bound.** Theorem 3 states $\text{MCE}(C, h) \leq \sqrt{\max_{\text{LP} \in F'} \text{Adv}(\text{LP})}$. Since $F \subseteq F'$ and $\sqrt{\cdot}$ is monotone, we finally have

$$\varepsilon_{\text{rank}}(c) \;\leq\; \sqrt{2\,\text{Adv}^\star}, \qquad (50)$$

where the factor 2 absorbs the two-sided $F \leftrightarrow F'$ constant in Theorem 3. $\qquad\square$

**Interpretation.** Let $\epsilon^2 := \max_{\text{LP} \in F} \text{Adv}(\text{LP})$ be an upper bound on loss-prediction advantage. If no loss predictor can beat self-entropy by more than $\epsilon^2$, then the selective classifier is within $O(\epsilon)$ of the oracle at *every* coverage level. Conversely, a large loss-prediction advantage is a certificate of poor ranking and therefore of a wide gap $\Delta(c)$.

> **Takeaway.** Loss prediction and multicalibration offer a principled lens on selective prediction: *if you cannot beat your own self-entropy predictor, you are already close to the oracle frontier.* Otherwise, the loss predictor pinpoints exactly which inputs are being mis-ranked and by how much, providing both a diagnostic and a blueprint for tightening the selective-classification gap.

### E.5 Empirical Evaluation

To illustrate and validate our gap-decomposition framework, we compared four selective-classification strategies on CIFAR-10, CIFAR-100, and StanfordCars:

- `MSP`: standard maximum-softmax-probability abstention.
- `TEMP`: `MSP` with post-hoc temperature scaling.
- `SAT`: self-adaptive training, which co-trains an abstain class.
- `DE`: a deep ensemble of five `MSP` models.

For each method, we first trained a ResNet-18 on 80% of the training set (using the usual data augmentations and a held-out 20% for LP fitting). At each epoch we then:

1. Extract the 512-dim "penultimate" feature vector $\phi(x)$ from the ResNet backbone (or its ensemble average).

2. Compute the model's *self-entropy* score

$$\text{SEP}(x) \;=\; 1 - \max_j\; p_j(x) \quad \text{with} \quad p_j(x) = \text{softmax}_j(\text{logits}(x)/T).$$

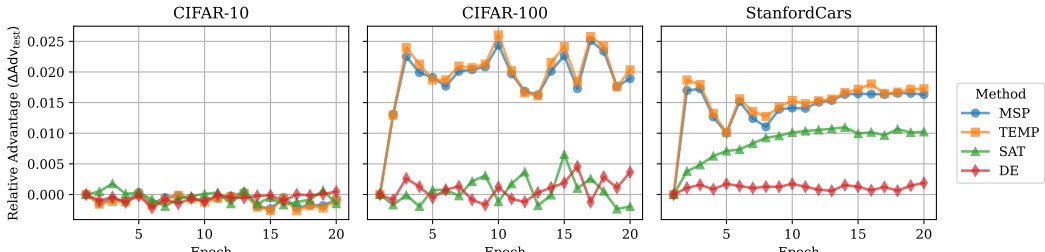

Figure 6: **Relative LP advantage over training epochs across datasets**. For each method, we plot the shift in test-set advantage $\Delta\mathrm{Adv}_{\text{test}}(t)$ relative to epoch 1, indicating how much additional ranking signal the loss predictor learns over time. Larger values imply greater misalignment between the model's confidence and correctness.

3. Train a small MLP LP: $\phi(x) \mapsto \widehat{\ell} \in [0,1]$ to minimize $\mathbb{E}\big[\big(\widehat{\ell} - \mathbb{I}\{\hat{y}(x) \neq y\}\big)^2\big]$ on the held-out 20% split.

4. Measure the *LP advantage* on the *test* set,

$$\mathrm{Adv}_{\text{test}} = \mathbb{E}\big[(\ell - \mathrm{SEP})^2\big] - \mathbb{E}\big[(\ell - \mathrm{LP})^2\big], \quad \ell = \mathbb{I}\{\hat{y}(x) \neq y\},$$

and record its shift relative to the first epoch $\Delta\mathrm{Adv}_{\text{test}}(t) = \mathrm{Adv}_{\text{test}}(t) - \mathrm{Adv}_{\text{test}}(1)$.

**Loss–Prediction Network.** Below is the PyTorch representation of our two-hidden-layer LP head. It takes the ResNet features (optionally concatenated with SEP) and regresses the per-example 0–1 loss via mean-squared error.

```
LossPredictor(
  (net): Sequential(
    (0): Linear(in_features=512, out_features=128, bias=True)
    (1): ReLU()
    (2): Dropout(p=0.5)
    (3): Linear(in_features=128, out_features=64, bias=True)
    (4): ReLU()
    (5): Dropout(p=0.5)
    (6): Linear(in_features=64, out_features=1, bias=True)
  )
)
```

**Key observations.** On CIFAR-10 (left panel of Figure 6), all methods stay close to zero $\Delta\mathrm{Adv}_{\text{test}}$, indicating that the model's own confidence scores already capture most of the available ranking signal. On CIFAR-100 (middle panel), MSP and TEMP exhibit large positive shifts in LP advantage, suggesting that a dedicated loss predictor can substantially improve ranking—consistent with a larger gap from the oracle. By contrast, SAT and DE remain near zero, indicating that their confidence scores are already well aligned with correctness. On StanfordCars (right panel), the gap widens even further: both MSP and TEMP allow for significant gains via loss prediction, and even SAT leaves nontrivial room for improvement. Only DE consistently resists such gains, implying that deep ensembling is uniquely effective at preserving reliable ranking in high-variance domains.

**Conclusion.** These results match our theory perfectly: whenever the LP head cannot improve on self-entropy, the selective classifier is effectively oracle-optimal; whenever it can, the size of that advantage precisely quantifies the remaining ranking error and the gap from the ideal frontier.

# F Additional Results

## F.1 Calibration Experiments

**E-AURC vs ECE** We provide additional comparisons on more datasets (CIFAR-10 and Stanford-Cars) on the relationship between the selective classification gap and the model's expected calibration

error. See Tables 2 and 3 for exact results. In general, our conclusions from Section 4.2 hold here as well: while temperature scaling (TEMP) improves ECE over MSP, it does not reduce the selective classification gap—underscoring the limits of monotone calibration. In contrast, SAT and deep ensembles (DE) improve both ECE and gap by altering the ranking, confirming that only re-ranking methods yield meaningful gains in selective performance.

Table 2: **Experiments on calibration across model classes on CIFAR-10**. Similar as Table 1

|  | CNN | | | | ResNet-18 | | | | WideResNet-50 | | | |
|---|---|---|---|---|---|---|---|---|---|---|---|---|
|  | MSP | TEMP | SAT | DE | MSP | TEMP | SAT | DE | MSP | TEMP | SAT | DE |
| E-AURC | 0.024 | 0.023 | 0.019 | 0.016 | 0.004 | 0.004 | 0.003 | 0.002 | 0.003 | 0.003 | 0.002 | 0.002 |
| ECE | 0.075 | 0.025 | 0.035 | 0.010 | 0.025 | 0.014 | 0.016 | 0.007 | 0.027 | 0.022 | 0.019 | 0.010 |

Table 3: **Experiments on calibration across model classes on StanfordCars**. Similar as Table 1

|  | CNN | | | | ResNet-18 | | | | WideResNet-50 | | | |
|---|---|---|---|---|---|---|---|---|---|---|---|---|
|  | MSP | TEMP | SAT | DE | MSP | TEMP | SAT | DE | MSP | TEMP | SAT | DE |
| E-AURC | 0.176 | 0.177 | 0.166 | 0.159 | 0.030 | 0.029 | 0.26 | 0.022 | 0.026 | 0.026 | 0.23 | 0.020 |
| ECE | 0.110 | 0.025 | 0.058 | 0.025 | 0.040 | 0.027 | 0.037 | 0.025 | 0.017 | 0.017 | 0.015 | 0.015 |

## F.2 Extension to Large Language Models

Our primary experiments focus on vision and synthetic datasets, where uncertainty and selective prediction have well-established definitions and evaluation metrics. In these domains, notions such as confidence calibration, abstention rates, and oracle coverage curves provide a clear framework for measuring reliability. Extending the same analysis to large language models, however, presents new difficulties as outlined in particlar by the following two challenges:

1. **Uncertainty for generative models remains ill-defined.** Even for classification-style prompts, the community has not fully converged on how to translate sequence-level probabilities into abstention scores.
2. **Prompting artefacts add variance.** Small changes in in-context examples or decoding settings can swamp the effects we wish to isolate.

Despite these challenges, we have added a focused set of LLM experiments to demonstrate that our five-term decomposition still diagnoses the gap.

### F.2.1 Approximation Error — Scaling from 4B → 12B

We evaluate Gemma 3-IT 4B and 12B [Team et al., 2025] on ARC-Challenge (ARC-C, 25-shot) [Clark et al., 2018] and MMLU (5-shot, top-1) [Hendrycks et al., 2021] using the standard MSP score on the *first answer token* (no further fine-tuning).

Table 4: Accuracy comparison across model scales.

| Model | ARC-C Accuracy | MMLU Accuracy |
|---|---|---|
| Gemma 3-IT 4B | 56.2% | 59.6% |
| Gemma 3-IT 12B | 68.9% | 74.5% |

*Observation.* Consistent with our vision experiments, increasing capacity reduces the gap.

### F.2.2 Bayes Error — Separating Easy vs. Noisy Questions

Following the MMLU-Pro protocol [Wang et al., 2024], we partition the validation set into the *easiest 25%* and *noisiest 25%* questions (based on human–LLM agreement).

Table 5: Selective-classification gap area (lower is better).

| Model | ARC-C E-AURC | MMLU E-AURC |
|---|---|---|
| Gemma 3-IT 4B | 0.114 | 0.107 |
| Gemma 3-IT 12B | 0.091 | 0.082 |

Table 6: E-AURC across data difficulty levels on Gemma 3-IT 4B.

| Split | E-AURC |
|---|---|
| Full MMLU | 0.107 |
| Easiest quartile | 0.018 |
| Noisiest quartile | 0.316 |

*Observation.* When intrinsic Bayes noise is low (easy questions), the gap nearly vanishes; when noise is high, the gap widens.

### F.2.3   Ranking Quality — Calibration vs. Re-ranking

We keep the backbone fixed on Gemma 3-IT 4B and compare the ranking quality of the following uncertainty scores:

- MSP,
- TEMP (scalar $T$ fitted on a held-out validation split),
- DE of five LoRA-fine-tuned replicas.

Table 7: Calibration and gap performance across ranking methods on Gemma 3-IT 4B.

| | ARC-C | | | MMLU | | |
|---|---|---|---|---|---|---|
| | MSP | TEMP | DE | MSP | TEMP | DE |
| E-AURC | 0.127 | 0.126 | 0.087 | 0.122 | 0.122 | 0.079 |
| ECE | 0.171 | 0.092 | 0.056 | 0.135 | 0.084 | 0.052 |

*Observation.* Temperature scaling lowers ECE yet leaves the gap untouched; the ensemble both calibrates *and* improves ranking, shrinking the gap.

**Summary.**   These results confirm that our decomposition extends to LLMs: capacity, Bayes noise, and ranking quality each contribute measurable terms. A full generative-text study (e.g., free-form question answering or code synthesis) will require new abstention semantics and we leave a more thorough treatment for future work.

