# OpenReview forum: "What Does It Take to Build a Performant Selective Classifier?"
_NeurIPS.cc/2025/Conference — NeurIPS 2025 poster_

### Official Review · Reviewer_7Yt4 · 2025-06-05

**Clarity:** 3
**Significance:** 3
**Originality:** 3
**Rating:** 4
**Confidence:** 4

**Summary:**

This paper presents an analytical framework for selective classification that decomposes the gap between a selective classifier and the optimal "oracle" into a number of well-defined components:

1. Bayes (Aleatoric uncertainty) resulting in natural label ambiguity.
2. Approximation error of the model to the true conditional distribution.
3. Ranking error of the confidence score.
4. Statistical error from using empirical estimates of expectations
5. Miscellaneous covering "the rest" such as optimisation imperfections.

The authors then demonstrate empirically the impact of each component on model performance.

Throughout the paper, the authors aim to provide practical takeaways to the reader for improving the gap.

**Questions:**

See weaknesses, I am keen to raise my score if the authors are able to sufficiently address the issues I've presented above. In fact, I am generally positive on the paper at a high level, and would like to see it improve.

**Ethical Concerns:**

["NO or VERY MINOR ethics concerns only"]

**Final Justification:**

(copied from another comment in the review discussion thread)

After rereading my discussion with the authors as well as their discussion with other reviewers, I would like to summarise my thoughts on the submission.

- My views on the strengths of the paper remain unchanged from my initial review.
- The authors (for the most part) have "patched" the various issues mentioned by myself and other reviewers, as well as taken in feedback relating to paper presentation/communication. I am appreciative of their efforts as the discussion has been extensive.

As such, **I do not have a strong reason to reject the submission**; however, due to the *extensive changes* undergone during rebuttal, the inability for reviewers to clearly view the current (post-rebuttal) state of the submitted manuscript, and the issues that exist in the original manuscript if viewed as is, I **strongly believe the paper would benefit from an additional round of peer review** (in agreement with reviewer 9vHG). This would allow the manuscript to be properly evaluated in its current, evolved state, and potentially give it the chance to receive a higher recommendation (I reiterate my positive view on the *potential* of the paper).

As such I will **raise my score to borderline accept**, however, I **urge the AC to consider the above when making their final recommendation**.

**Limitations:**

Yes

**Quality:**

2

**Strengths And Weaknesses:**

## Strengths

- The knowledge presented could be of practical use to a range of readers and practictioners, facilitating better real-world systems for selective classification. I think there is real value in being able to disentangle the contributing causes to a problem in order to build better systems in a more informed way. For example, in existing research aimed at improving selective classification, especially *training* based approaches, it is not always clear whether improved selection is due to better ranking (as is typically how methods are motivated) or actually simply due to better accuracy due to improved generalisation performance.
- Attractive and clear presentation. The authors structure the paper in a way that is easy to follow, with important information visually highlighted.
- Good balance of theory and practical information. Although the paper is theory heavy, practical takeaways are presented intuitively in natural language, reducing the barrier-to-entry for the reader. Thumbs up especially for the inclusion of Appendix C.

## Weaknesses

As a reviewer, I can certainly make mistakes. Due to the large reviewing load for NeurIPS I may have missed information in the paper, or misinterpreted details. I welcome the authors to correct me if that is the case.


1. **Missing nuance in the presentation of the oracle**: I believe there are some presentation issues here
    1. It is not made clear in Definition 3 that the upper bound is achieved under no Bayes noise.
    2. It is not clear whether the oracle is the bayes-optimal ranker (in the presence of Bayes noise) or a ranker with "oracle" knowledge of the true sampled label and thus the correctness of the classifier.
    3. It is not made clear that lower approximation error (higher mean accuracy $a_\text{full}$) also raises the upper bound rather than just decreasing the selective gap. How this interacts with Eqs. 9-12 is also not discussed. In fact, this is clearly shown in Fig. 2 but not emphasised.
2. **Overstated contribution on monotone calibration**. The fact that *monotone calibration cannot improve selective classification* should indeed be emphasised, but to claim this result as a contribution is odd to me as it seems to obviously arise from basic knowledge of binary classification. I would recommend the authors temper this claimed contribution.
3. **Space for improvement presenting Eqs. 9-12**. Although I have previously praised the presentation, I feel like Sec. 3.3 would benefit from guiding the reader a bit more. For example:
    1. Making it clear that Eq. 10 compares the learned classifier to the optimal classifier, *whilst keeping the selection function constant*. Note that this is not necessarily intuitive as in practice changing the classifier generally leads to changing the selection function.
    2. Bringing in discussion (mentioned in the limitations section) about how a single design choice might affect multiple components. E.g. Typically, MSP is used by default, so improving the approximation error of the model $P_\theta(y|x)$ to the true conditional will simultaneously improve both $\varepsilon_\text{rank}$ and $\varepsilon_\text{approx}$. This sort of clarification would be especially useful to practitioners.
4. **Numerous issues with calibration discussion**
    1. The claim that *temperature scaling rarely reorders* is supported by incorrect calculations in Appendix B.2 (simply sanity checking the provided logits in Eq. 34 gives different softmax outputs, and Eq. 32 seems to be wrong). Besides, existing research [1,2] (mentioned by the authors) clearly demonstrates empirically that temperature scaling has a large effect on selective classification.
    2. Poor discussion on scoring. Histogram binning, which explicitly introduces ties by many-to-one mappings, is claimed to preserve ordering (even though ties are later discussed in the paper). Recent work [2,3] that both empirically demonstrates and analytically explains the effectiveness of post-training logit normalisation for improving selection, as well as investigates the effects of the calibration approach label smoothing, are not discussed.
5. **Various missing discussion**. I believe a number of recent research directions should be at least briefly discussed as being relevant (e.g. for future work).
    1. Unified selective classification [4,5,6] including OOD data to be rejected. How could the gap be extended in this case?
    2. Uncertainty for open-ended language generation [7,8]. What challenges exist as our models are no longer directly trained to model the Bayes posterior?
7. **Advice not backed by empirical evidence**. Although I have praised the presence of Appendix C, much of the advice given is not based on experiments, but simply the authors' understanding of existing approaches. This could lead to potential issues. For example, [3] explains how label smoothing, an approach that superficially should improve both approximation error and calibration, in fact degrades selective classification. Additionally, they recommend binning approaches, even though binning introduces ties.  Authors ought to roll back advice or provide additional experiments/references to other works with results for specifically selective classification, to support the advice given.
6. **Minor complaints**.
    1. The use of colour, although aesthetic, is a little arbitrary. I initially was confused as I thought definitions and takeaways were the same colour.
    2. Sec. 2 Accuracy-coverage tradeoff is missing a reference [9].
    3. Both MSP and SR are used to refer to the same thing.

**Note**: I do not see the paper's lack of a novel method/algorithm as a weakness, as I believe this sort of framework/analysis paper can be just as, if not more impactful than a paper that proposes new solutions.

[1] Galil et al. What Can we Learn From The Selective Prediction And Uncertainty Estimation Performance Of 523 Imagenet Classifiers? *ICLR 2023*

[2] Cattelan and Silva How to Fix a Broken Confidence Estimator: Evaluating Post-hoc Methods for Selective Classification with Deep Neural Networks *UAI 2024*

[3] Xia et al. Towards Understanding Why Label Smoothing Degrades Selective Classification and How to Fix It *ICLR 2025*

[4] Jaeger et al. A Call to Reflect on Evaluation Practices for Failure Detection in Image Classification *ICLR 2023*

[5] Xia and Bouganis Augmenting the Softmax with Additional Confidence Scores for Improved Selective Classification with Out-of-Distribution Data *IJCV 2024*

[6] Narasimhan et al. Plugin estimators for selective classification with out-of-distribution detection *ICLR 2024*

[7] Kuhn et al. Semantic Uncertainty: Linguistic Invariances for Uncertainty Estimation in Natural Language Generation *ICLR 2023*

[8] Kang et al. Scalable Best-of-N Selection for Large Language Models via Self-Certainty 2025

[9] Traub et al. Overcoming Common Flaws in the Evaluation of Selective Classification Systems *NeurIPS 2024*

---

> ### Author Rebuttal · Authors · 2025-07-29
>
> We thank the reviewer for their in-depth feedback, and for **highlighting the clarity, practical value, and balance of our paper**. The reviewer raises important points about the **presentation of the oracle bound, the calibration discussion, and the need for clearer exposition and empirical support in parts**. We address each of these concerns below and have revised the paper to incorporate the suggested corrections.
>
> ---
>
> ``1) Missing nuance in the presentation of the oracle.``
>
> ``Definition 3 assumes no Bayes noise.``
>
> We agree with the reviewer that this assumption was indeed not stated explicitly. We have revised the sentence immediately following Definition 3:
>
> > Assuming no Bayes noise—that is, all errors are avoidable given perfect confidence—this piecewise curve (see Figure 1) traces the oracle accuracy–coverage frontier based on a perfect ranking of examples by correctness probability.
>
> ``What kind of oracle?``
>
> After rephrasing the previous sentence, this distinction becomes immediate under the no-Bayes-noise assumption: when every example’s posterior correctness $\Pr\bigl(h(x)=Y \mid x\bigr)\in\{0,1\}$, ranking by the Bayes-optimal posterior is identical to ranking by the true realized label. Hence, **in the no-Bayes-noise setting the two oracle interpretations coincide**.
>
> ``State that lower approximation error also raises the upper bound.``
>
> As Definition 3 makes explicit, the oracle curve $\overline{\mathrm{acc}}(a_{\text{full}}, c)$ depends directly on the full-coverage accuracy $a_{\text{full}}$. Thus, improving approximation error (i.e., raising $a_{\text{full}}$) not only reduces the selective classification gap $\Delta(c)$, but also lifts the upper bound itself across all coverage levels. This interaction is indeed visible in Figure 2 but was indeed not emphasized in the text. **We have now clarified this point directly after Definition 4** and revised the figure caption to highlight how changes in approximation accuracy affect both the bound and the observed gap.
>
> ---
>
> ``2) Overstated contribution on monotone calibration.``
>
> We thank the reviewer for pointing out this issue and have revised the draft to clarify our intent. While it is well understood that monotone transforms preserve score ordering and therefore cannot improve ranking-based risk–coverage curves, **this point is often muddled in the literature**—especially given empirical studies [1, 2] that appear to show large selective-classification gains from temperature scaling. Our contribution is to **(i)** formalize this invariance result in the multi-class case, **(ii)** identify rare logit-margin ties as the only way monotonicity can fail to preserve ranking, and **(iii)** highlight that performance improvements must stem from calibration methods that actively rerank examples. We now present this section as a clarification **aimed at resolving apparent contradictions in prior work**, not as a standalone result. We hope this helps readers more clearly distinguish between calibration techniques that change confidence scores and those that change rank order.
>
> ---
>
> ``3) Space for improvement presenting Eqs. 9-12``
>
> ``Clarify that Eq. 10 keeps the selection function constant.``
>
> Thank you for pointing out the potential ambiguity. To make the role of each term crystal-clear, we have **added an additional clarification** on this point after Eq. (9)-(12) that explicitly states:
>
> > Here we freeze the acceptance set $A_c$ defined by the current scoring function and ask how much worse the learned classifier $h$ is than the Bayes-optimal rule on this *same* set.
>
> We then remind the reader that any shifts in the acceptance region caused by replacing $h$ are *not* charged to $\varepsilon_{\text{approx}}(c)$ but are instead captured by the ranking-error term $\varepsilon_{\text{rank}}(c)$ in Eq. (11). This clarification makes it evident that the two terms partition the total gap without double-counting.
>
> ``Foreshadow discussion about how a single design choice might affect multiple components.``
>
> This is a great suggestion. We have now **added a short “cross-term interactions” paragraph** at the end of §3.3 that makes this explicit.
>
> > **A single design choice can shrink multiple error terms.** When the confidence score is the maximum softmax probability (MSP), a better approximation of the true conditional $\eta$ not only lowers the *approximation* term $\varepsilon_{\text{approx}}(c)$ but also tends to align MSP more closely with $\eta_h$, thereby *indirectly reducing* the ranking error $\varepsilon_{\text{rank}}(c)$. Conversely, a non-monotone calibration head can slash $\varepsilon_{\text{rank}}(c)$ without improving $\varepsilon_{\text{approx}}(c)$.
>
> We believe this forward-reference clarifies how model-capacity upgrades, richer pre-training, or loss-function tweaks can yield compound benefits, and we direct readers to the limitations section for a fuller discussion of such interactions.
>
> ---
>
> ``4) Numerous issues with calibration discussion.``
>
> `` Incorrect calculations in Appendix B.2.``
>
> We thank the reviewer for catching the errors in Appendix B.2. We have **thoroughly revised this section to correct both the analytic expression for temperature-scaled confidence scores and the illustrative example**. The previous equation for $s_T(x)$ incorrectly assumed a binary logistic form; we now provide the correct expression for arbitrary $K$:
>
> $$
> s_T(x) = \frac{1}{1 + \sum_{j \ne j_\star} \exp[(z_j - z_{j_\star})/T]} = \frac{1}{1 + \sum_{j \ne j_\star} r_j^{1/T}}.
> $$
>
> We also replaced the earlier example, which wrongly claimed a ranking swap, with corrected logits that do exhibit a swap at higher temperature: $s_1(x_1) = 0.576 > 0.512 = s_1(x_2)$ at $T = 1$, but $s_3(x_1) = 0.411 < 0.428 = s_3(x_2)$ at $T = 3$. Finally, we clarified the condition for reordering: that the sums over non-max logits, $S_1(T) = S_2(T)$, cross for some $T$.
>
> ``Existing research [1,2] demonstrates that TEMP has a large effect on SC.``
>
> We thank the reviewer for pointing to prior work suggesting that temperature scaling (TS) improves selective prediction. **The discrepancy arises from differences in evaluation setup**. Our analysis (§3.4, App. B.2) focuses on ranking-based selective classifiers, where prediction is accepted based on the top-\$c\$ fraction ranked by \$s(x)\$. In this setting, monotonic transformations like TS preserve the ordering and thus do not affect the selective classification gap \$\Delta(c)\$ or selective accuracy at fixed coverage. In contrast, [1] uses a fixed score threshold \$\tau\$ (e.g., “accept if \$s(x) > \tau\$”), where TS can shift scores so that more correct predictions exceed \$\tau\$, improving metrics like AUROC. However, **AUROC mixes overall accuracy and ranking quality**, whereas our decomposition isolates the selective component by conditioning on rejection. As shown in §4.2, while TS may improve calibration, it does not affect \$\Delta(c)\$ unless it changes the ranking. We have clarified this in the revised draft.
>
> ``Poor discussion on scoring/binning.``
>
> We appreciate the reviewer’s observation and agree that our original discussion of histogram binning was too brief. While histogram binning fits a monotone mapping, it does induce score ties by mapping wide score ranges to the same value; **we now clarify in §3.4 and Appendix B.2 how such ties can—and in rare cases do—affect the selection ordering**. We have also added reference to the mentioned works [2,3] on post-training logit normalization and label smoothing.
>
> ---
>
> ``5) Discussion on future directions, including (i) connections to OOD; and (ii) uncertainty in open-ended generative (sequence) models.``
>
> We appreciate the reviewer’s pointer to several fast-moving research threads that lie just beyond the scope of our current study. While we think that these directions should be more thoroughly investigated in future work, we want to acknowledge these connections. Hence, we have **created a new Appendix F** (“Beyond the Closed-World Setting”) that provides two self-contained subsections (§F.1 and §F.2) outlining how our decomposition might be generalized and which technical obstacles remain. We also include the following summary and pointer in our conclusion section:
>
> > Our finite-sample gap decomposition lays the groundwork for a more unified reliability framework; extending it to (i) settings where out-of-distribution inputs must be rejected and (ii) open-ended language generation—where correctness is semantic rather than categorical—constitutes a promising agenda for future work that we outline in Appendix F.
>
> We are happy to provide a more detailed overview of these sections upon the reviewer’s request.
>
> ---
>
> ``6) Advice given in Appendix C.``
>
> We have clarified in Appendix C **which checklist items derive from our theoretical and empirical results versus established best practices**, and we’ve **supported each recommendation with appropriate citations** (including the reviewer’s suggestions). To avoid confusion, we now explicitly note [3] cautioning against the use of label smoothing (which notably was not part of our original recommendations). For techniques with mixed effects—such as binning—we cross‐reference tie‐breaking remedies and advise tuning bin granularity to prevent coverage cliffs.
>
> ---
>
> ``7) Minor issues: Definition vs takeaways color; Sec. 2 accuracy-coverage tradeoff reference; MSP and SR are both used.``
>
> We thank the reviewer for bringing these issues to our attention. We have adjusted the colors to be more distinctive between definitions and take-aways, have added a reference to [9], and have replaced the single mention of SR in §2 with MSP.
>
> ---
>
> **We hope that our rebuttal has addressed the reviewer’s questions and concerns and that the reviewer considers raising their individual and overall score(s). Our work has already significantly improved thanks to their feedback. We are happy to further engage with them during the discussion period!**

---

> ### Comment · Reviewer_7Yt4 · 2025-08-03
>
> I thank the authors for their clarifications, as well as for their incorporation of my feedback. I apologise for the delayed reply, I have been travelling over the weekend. There are still a few aspects of their response that I am unsure about however.
>
> 1. Temperature scaling does not change the ranking of logits *for a given sample* over classes, keeping accuracy@100% coverage *constant* for a given model, and therefore the number of correct and incorrect samples constant. Galil et al [1] measure the change in AUROC before and after temperature scaling for each classifier, and demonstrate meaningful changes. As the number of in/correct samples doesn't change, AUROC is a proxy for ranking/binary classification performance, and will only change if the confidence ranking between samples changes. Besides, Cattelan and Silva [2] demonstrate TS has a large effect on their metric, normalized AURC for various individual classifiers. This is empirical evidence over *hundreds* of models that temperature scaling changes ranking.
> 2. I firmly believe that histogram binning should not be recommended if the objective is to improve selective classification. Typically the order of 10 bins is used -- quantising each bin effectively randomises all confidence scores within each bin, reducing intra bin discrimination to random. Table 4 in [10]'s appendix demonstrates how *histogram binning always degrades selective classification*.
> 3. I am wary about Appendix C, could the authors provide concretely which advice they have given in the revised version and the supporting references? After all [1,2] have demonstrated that optimising for top 1 accuracy may severely negatively impact ranking performance -- many of their worst-performing models use the choices suggested in the Shrink Approximation Error section. Moreover, [10] demonstrate in their Table 4 that vector/Dirichlet scaling does not give consistent improvements and often hurts ranking performance.
>
> [10] Le Coz et al. Confidence Calibration of Classifiers with Many Classes *NeurIPS 2024*

---

> > ### Author Response · Authors · 2025-08-06
> > **Response (1/4)**
> >
> > We appreciate the reviewer’s active involvement in the discussion phase and in turn also apologize for our delay in responding to the reviewer. We wanted to make sure that we post our response at the same time as our reply to reviewer 9vHG.
> >
> > ---
> >
> > ``1) Temperature scaling improvements to ranking.``
> >
> > The reviewer correctly points out that **TS can change the ranking of samples**, which can lead to improved selective classification performance. We would like to clarify that **our paper already acknowledges this possibility**. However, we concede that our original phrasing may have understated the practical impact of this effect. To address the reviewer's valid concerns, we have revised the manuscript to state more clearly that TS can lead to modest but noticeable improvements in ranking, as empirically demonstrated in the work by Galil et al. [1] and Cattelan and Silva [2].
> >
> > Our **central argument, however, remains unchanged and is in fact strengthened by this clarification**. While TS can improve ranking, its effect is fundamentally limited because **it is a post-hoc method operating only on the model's output logits**. The primary contribution of our work is to provide a formal decomposition of the selective classification gap, which isolates the ranking error ($\varepsilon_\text{rank}$). This framework allows us to show that **methods designed to improve uncertainty and correctness prediction during training or through ensembling (like SAT and DE) are substantially more effective at minimizing this specific error term than TS**. Our novel results connecting strong ranking to loss prediction further underscore this point, demonstrating a clear performance hierarchy where DE and SAT significantly outperform TS-based approaches.
> >
> > In summary, we are not claiming TS has no effect; rather, we are providing a framework that explains why its effect on ranking is limited and why other methods are superior. **We have revised our paper to make this distinction clearer and to better contextualize our work with the papers the reviewer cited**. Here are our concrete changes:
> >
> > - New paragraph title in line 167:
> > > The Effect of Temperature Scaling on the SC Gap.
> > - Updated lines 169-172:
> > > As we note, temperature scaling can change the ranking of samples by confidence because the softmax function is non-linear with respect to the temperature parameter. This re-ranking can lead to modest but empirically validated improvements in selective classification performance, as measured by metrics like AUROC [1, 2]. However, the magnitude of this effect is inherently limited (see Appendix B.2 for an extended discussion.). While TS can refine the ordering, it does not fundamentally alter the underlying quality of the model's uncertainty estimates.
> > - Updated take-away box:
> > > While post-hoc calibration with temperature scaling can provide modest improvements to ranking, it is not sufficient to close the selective classification gap. Substantially reducing the ranking error ($\varepsilon_\text{rank}$) requires more powerful scoring methods that actively re-rank examples based on richer information, such as feature-aware heads, ensembles, or non-monotone transformations.
> >
> > ---
> >
> > ``2) The effect of binning.``
> >
> > We apologize for initially misunderstanding the reviewer’s concern in their original review and thank them for alerting us to [10] which we unfortunately were not aware of at the time of paper submission. **After carefully examining this work, in particular Appendix G, we agree with the reviewer that binning, alongside with vector/Dirichlet scaling should not be recommended in our work.** We have made sure to remove this mention from the methods section and have updated our checklist accordingly. See our updates to our method section below:
> >
> > > **Moving the gap requires non-monotone scoring.** To reduce the selective classification gap $\Delta(c)$, it is not enough to calibrate scores post-hoc using monotone mappings. One must actively change the ranking of accepted examples to better reflect true correctness likelihood. This often requires scoring mechanisms that incorporate instance-specific information—such as deep ensembles (DE), self-adaptive training (SAT), or learned correctness heads $g_\psi(x)$ that map hidden representations to confidence estimates. These approaches leverage model diversity or internal feature structure to separate samples that would otherwise receive indistinguishable confidence scores under standard softmax outputs.
> >
> > *(continued in next reply)*

---

> > > ### Author Response · Authors · 2025-08-06
> > > **Response (2/4)**
> > >
> > > > **Why binning and vector scaling should not be used.** Histogram binning and vector/Dirichlet scaling—while widely to improve calibration—are poorly suited for selective classification. Histogram binning quantizes scores into a small number of bins, mapping wide score intervals to the same value and destroying within-bin ordering, which leads to effectively random selection among tied examples. Vector and Dirichlet scaling are post-hoc calibration methods that generalize temperature scaling by learning class-specific transformations of logits—vector scaling applies a linear transformation, while Dirichlet scaling interprets the logits as parameters of a Dirichlet distribution to better model uncertainty. Recent work by Le Coz et al. [2024] shows that histogram binning and vector/Dirichlet scaling consistently degrade AUROC in selective classification, and even their proposed Top-versus-All (TvA) method—while improving calibration—does not enhance, and sometimes worsens, selective performance. These results underscore our central claim: improving calibration does not guarantee better ranking. Reducing the selective classification gap requires score functions that explicitly learn to separate easy from hard examples, not just to produce better-calibrated probabilities.
> > >
> > > ---
> > >
> > > ``3) Show revised version of checklist with references``
> > >
> > > Unfortunately, due to the NeurIPS policy of not allowing updates to the paper draft during the discussion phase, we are unable to provide the exact updated checklist as it appears in the paper. We include our best effort to reproduce our updated Appendix C in Markdown below. Assembling this newly updated checklist was a significant amount of work which we hope the reviewer will appreciate:
> > >
> > > ### $\varepsilon_{\text{approx}}$ — Shrink Approximation Error
> > >
> > > - **Model capacity:** Upgrade to deeper or wider architectures like ResNeXt, ViT, or ConvNeXt to better approximate complex functions and reduce base error [1,2,3,4].
> > > - **Pre-training:** Initializing with rich features from self-supervised methods (SimCLR, BYOL) or foundation models (CLIP, DINO) can improve out-of-the-box performance, convergence, and uncertainty scores [5,6,7,8,9]. However, pre-training can also sometimes negatively affectt selective classification performance [10].
> > > - **Distillation:** Use teacher–student training with logit matching or feature hints to inherit accuracy from a larger model at lower cost [10,11,12].
> > > - **Data augmentation:** Augmentations can often improve generalization with policy-based (AutoAugment, RandAugment) or mixing-based (MixUp, CutMix) augmentations to regularize the learner [13,14,15,16]. However, strong augmentations may also degrade selective classification performance for certain minority classes [17].
> > >
> > > ### $\varepsilon_{\text{rank}}$ — Improve Ranking
> > >
> > > - **Feature-aware scoring:** Train auxiliary heads like ConfidNet to learn correctness scores using both logits and input features [18], often improving uncertainty estimates. Self-Adaptive Training (SAT) further enhances this by encouraging internal representations to separate correct and incorrect predictions through contrastive regularization or supervised signals [19].
> > > - **Deep ensembles**: Use the disagreement or predictive entropy across multiple independently trained models to estimate uncertainty [20].
> > > - **Conformal methods:** Generate conformal p-values or risk-controlled selection sets that respect desired coverage guarantees [21,22].
> > > - **Use caution with vector/Dirichlet scaling:** While previous work has shown that vector, matrix, or Dirichlet transformations can be beneficial to reshape confidence distributions [23,24], [25] shows that these techniques can harm ranking under a large number of classes.
> > >
> > > ### $\varepsilon_{\text{opt}}$ — Reduce Optimization Error
> > >
> > > - **Convergence diagnostics:** Track training/validation loss curves to detect underfitting and determine optimal stopping points [26].
> > > - **Learning-rate schedules:** Employ dynamic LR strategies like cosine decay, OneCycle, or CLR to reach better optima more consistently [27,28,29].
> > > - **Checkpoint ensembles:** Save and average late-stage checkpoints or use snapshot ensembling to smooth optimization variance [30,20].
> > > - **Regularization:** Use dropout, weight decay, or stochastic depth to prevent overfitting and stabilize training [31,32,33].
> > >
> > > ### $\varepsilon_{\text{Bayes}}$ — Quantify Irreducible Noise
> > >
> > > - **Repeated labels:** Collect multiple annotations (e.g., CIFAR-10H) to estimate human-level disagreement and the Bayes error floor [34,35].
> > > - **Noise-robust training:** Mitigate label noise using bootstrapped or Taylor-truncated loss functions that temper reliance on hard labels [36,37].
> > > - **Dataset curation:** Apply confident learning to flag likely label errors or use active learning for relabeling efforts [38].
> > >
> > > *(continued in next reply)*

---

> > > > ### Author Response · Authors · 2025-08-06
> > > > **Response (3/4)**
> > > >
> > > > ### $\varepsilon_{\text{stat}}$ — Control Statistical Slack
> > > >
> > > > - **Validation set size:** Use a sufficiently large holdout set (typically 10–20%) to estimate thresholds and calibrate uncertainty reliably [39].
> > > > - **Confidence intervals:** Use DKW or Clopper–Pearson bounds to set conservative thresholds with statistical guarantees on coverage [40,41].
> > > > - **Cross-validation:** Average selection thresholds over folds to reduce their variance and avoid overfitting to a single validation set [42].
> > > >
> > > > ### $\varepsilon_{\text{shift}}$ — Mitigate Distribution Shift
> > > >
> > > > - **Shift detection:** Detect covariate shift via statistical two-sample tests such as MMD or KL divergence between feature distributions [43,44].
> > > > - **Importance weighting:** Correct mismatched data distributions with density ratio weighting, e.g., using kernel mean matching [45].
> > > > - **Domain adaptation:** Finetune with in-domain examples or use unsupervised techniques like AdaBN or domain-adversarial training (DANN) [46,47].
> > > > - **Test-time adaptation:** Adapt models at inference using entropy minimization (Tent) or batch norm recalibration to restore accuracy under shift [48,49].
> > > >
> > > > ### $\varepsilon_{\text{thr}}$ — Threshold–Selection Noise
> > > >
> > > > - **Bootstrap resampling:** Estimate variability in the selection threshold $\tau_c$ by computing its standard error across bootstrap samples [50].
> > > > - **Smooth thresholds:** Interpolate between adjacent scores or accept a random subset at the threshold to reduce coverage discontinuities [51].
> > > >
> > > > ### $\varepsilon_{\text{tie}}$ — Tie-Breaking & Score Quantization
> > > >
> > > > - **Use higher precision:** Use higher float precision (e.g., FP32 or FP64) or more logits bits to distinguish close scores and avoid ties [52].
> > > > - **Dithering:** Add tiny random noise to scores before thresholding to stochastically resolve ties and reduce instability.
> > > > - **Refrain from binning**: Histogram binning (HQ) or Bayesian Binning into
> > > > Quantiles (BBQ) often improve calibration but not selective classification performance [53,54,25].
> > > >
> > > >
> > > > ### References
> > > >
> > > > [1] Xie et al., "Aggregated Residual Transformations for Deep Neural Networks," CVPR, 2017.
> > > > [2] Dosovitskiy et al., "An Image is Worth 16x16 Words: Transformers for Image Recognition at Scale," ICLR, 2021.
> > > > [3] Liu et al., "A ConvNet for the 2020s," CVPR, 2022.
> > > > [4] Kadavath Saurav et al. "Language models (mostly) know what they know." arXiv, 2022.
> > > > [5] Chen et al., "A Simple Framework for Contrastive Learning of Visual Representations," ICML, 2020.
> > > > [6] Grill et al., "Bootstrap Your Own Latent: A New Approach to Self-Supervised Learning," NeurIPS, 2020.
> > > > [7] Radford et al., "Learning Transferable Visual Models From Natural Language Supervision," ICML, 2021.
> > > > [8] Caron et al., "Emerging Properties in Self-Supervised Vision Transformers," ICCV, 2021.
> > > > [9] Hendrycks et al., "Using pre-training can improve model robustness and uncertainty." ICML, 2019.
> > > > [10] Galil et al., "What Can We Learn From The Selective Prediction And Uncertainty Estimation Performance Of 523 Imagenet Classifiers?"" ICLR, 2023.
> > > > [11] Hinton et al., "Distilling the Knowledge in a Neural Network," arXiv, 2015.
> > > > [12] Romero et al., "FitNets: Hints for Thin Deep Nets," ICLR, 2015.
> > > > [13] Cubuk et al., "AutoAugment: Learning Augmentation Policies from Data," CVPR, 2019.
> > > > [14] Cubuk et al., "RandAugment: Practical Automated Data Augmentation," NeurIPS, 2020.
> > > > [15] Zhang et al., "MixUp: Beyond Empirical Risk Minimization," ICLR, 2018.
> > > > [16] Yun et al., "CutMix: Regularization Strategy to Train Strong Classifiers with Localizable Features," ICCV, 2019.
> > > > [17] Jones et al. "Selective classification can magnify disparities across groups." ICLR, 2021.
> > > > [18] Corbière et al., "Addressing Failure Prediction by Learning Model Confidence," NeurIPS, 2019.
> > > > [19] Huang et al. "Self-adaptive training: beyond empirical risk minimization." NeurIPS, 2020.
> > > > [20] Lakshminarayanan et al., "Simple and Scalable Predictive Uncertainty Estimation using Deep Ensembles," NeurIPS, 2017.
> > > > [21] Vovk et al., "Algorithmic Learning in a Random World," Springer, 2005.
> > > > [22] Angelopoulos et al., "Conformal Risk Control," ICML, 2023.
> > > > [23] Guo et al., "On Calibration of Modern Neural Networks," ICML, 2017.
> > > > [24] Kull et al., "Beyond Temperature Scaling: Obtaining Well-Calibrated Multi-Class Probabilities with Dirichlet Calibration," NeurIPS, 2019.
> > > > [25] Le Coz et al. "Confidence calibration of classifiers with many classes." NeurIPS, 2024.
> > > > [26] Goodfellow et al., "Deep Learning," MIT Press, 2016.
> > > > [27] Loshchilov & Hutter, "SGDR: Stochastic Gradient Descent with Warm Restarts," ICLR, 2017.
> > > > [28] Smith & Topin, "Super-Convergence: Very Fast Training of Neural Networks Using Large Learning Rates," arXiv, 2018.
> > > > [29] Smith, "Cyclical Learning Rates for Training Neural Networks," WACV, 2017.
> > > > [30] Huang et al., "Snapshot Ensembles: Train 1, Get M for Free," ICLR, 2017.
> > > >
> > > > *(continued in next reply)*

---

> > > > > ### Author Response · Authors · 2025-08-06
> > > > > **Response (4/4)**
> > > > >
> > > > > [31] Srivastava et al., "Dropout: A Simple Way to Prevent Neural Networks from Overfitting," JMLR, 2014.
> > > > > [32] Huang et al., "Deep Networks with Stochastic Depth," ECCV, 2016.
> > > > > [33] Loshchilov & Hutter, "Fixing Weight Decay Regularization in Adam," ICLR, 2019.
> > > > > [34] Peterson et al., "Human Uncertainty Makes Classification More Robust," ICML, 2019.
> > > > > [35] Wei et al., "Learning with Noisy Labels Revisited: A Study Using Real-World Human Annotations," NeurIPS, 2021.
> > > > > [36] Reed et al., "Training Deep Neural Networks on Noisy Labels with Bootstrapping," ICLR, 2015.
> > > > > [37] Feng et al., "Can Cross Entropy Loss Be Robust to Label Noise?" IJCAI, 2020.
> > > > > [38] Northcutt et al., "Confident Learning: Estimating Uncertainty in Dataset Labels," JMLR, 2021.
> > > > > [39] Duda et al., "Pattern Classification," Wiley, 2000.
> > > > > [40] Massart, "The Tight Constant in the Dvoretzky–Kiefer–Wolfowitz Inequality," Ann. Prob., 1990.
> > > > > [41] Clopper & Pearson, "The Use of Confidence or Fiducial Limits," Biometrika, 1934.
> > > > > [42] Kohavi, "A Study of Cross-Validation and Bootstrap for Accuracy Estimation," IJCAI, 1995.
> > > > > [43] Gretton et al., "A Kernel Two-Sample Test," JMLR, 2012.
> > > > > [44] Rabanser et al., "Failing Loudly: An Empirical Study of Methods for Detecting Dataset Shift," NeurIPS, 2019.
> > > > > [45] Huang et al., "Correcting Sample Selection Bias by Unlabeled Data," NIPS, 2006.
> > > > > [46] Ganin et al., "Domain-Adversarial Training of Neural Networks," JMLR, 2016.
> > > > > [47] Li et al., "Revisiting Batch Normalization for Practical Domain Adaptation," ICCV, 2016.
> > > > > [48] Nado et al., "Evaluating and Calibrating Uncertainty for Deep Learning," NeurIPS, 2021.
> > > > > [49] Wang et al., "Tent: Fully Test-Time Adaptation by Entropy Minimization," ICLR, 2021.
> > > > > [50] Efron & Tibshirani, "An Introduction to the Bootstrap," Chapman & Hall, 1993.
> > > > > [51] Angelopoulos et al., "A Gentle Introduction to Conformal Prediction and Distribution-Free Uncertainty Quantification," arXiv, 2021.
> > > > > [52] Micikevicius et al., "Mixed Precision Training," ICLR, 2018.
> > > > > [53] Naeini et al., "Obtaining Well Calibrated Probabilities Using Bayesian Binning," AAAI, 2015.
> > > > > [54] Gupta et al., "CALIBER: A Benchmark Suite for Calibration Evaluation," NeurIPS Datasets and Benchmarks, 2021.
> > > > >
> > > > > **We are happy to incorporate additional feedback/suggestions from both reviewer 7Yt4 and 9vHG into this checklist since they are clearly experts in the field of selective prediction. We greatly value both their inputs to improve our paper.**
> > > > >
> > > > > ---
> > > > >
> > > > > **We hope that these additional clarifications have resolved the reviewers outstanding concerns and remain available for further discussion.**

---

> > > > > > ### Comment · Reviewer_7Yt4 · 2025-08-06
> > > > > >
> > > > > > I thank the authors for receiving my feedback on binning and temperature scaling, as well as their effort in providing the updated checklist with citations. However, I still have remaining concerns.
> > > > > >
> > > > > > 1. [2,3] demonstrate clearly that in *certain cases* post-hoc logit adjustments can have a large effect on $\varepsilon_\text{rank}$. Consider Figure 1 in Cattelan and Silva [2], which shows a significant improvement for EfficientNet-V2-XL after post-training logit adjustment. [2,3] also note that these adjustments don't always help, in particular [3] show that models trained in a "vanilla" fashion without advanced data augmentations or label smoothing don't benefit from logit normalisation. Just because the experiments in the reviewed paper don't demonstrate post-hoc approaches having much of an effect, doesn't mean that this is always the case -- the literature shows that both cases of post-hoc approaches working and not working.
> > > > > >
> > > > > > 2. The authors recommend Mixup, however, I am suspicious of the effect of mixup on uncertainty estimation. For example [11] find that mixup degrades OOD detection, which is a related task to SC. Furthermore, many of the "broken" models discussed in [2] also include mixup in their training recipes. It is precisely this sort of uncertainty about side effects that worries me about the checklist. Although some recommendations are common sense, for others it is hard to be sure that they won't have unintended consequences as many have not been explicitly investigated in the context of SC.
> > > > > >
> > > > > > 3. The authors recommend ConfidNet and SAT for improving $\varepsilon_\text{rank}$, even though [12] explicitly empirically demonstrates that the improved SC of the above approaches actually arises from a reduction in $\varepsilon_\text{approx}$ and that discarding the selection head/logit and just using MSP actually gives better performance (see their Table 2).
> > > > > >
> > > > > > I further apologise for not being clearer with these ahead of time, as some of these papers are occurring to me as I am engaging in the review process.
> > > > > >
> > > > > > I am rather conflicted as I genuinely like this paper and appreciate the authors' efforts in incorporating the reviewers' feedback, however, I feel like there are currently too many issues present for it to be ready for publication. Moreover, since the submitted manuscript cannot be updated during review process there is sadly no way to view the authors' reported updates in context. I am leaning toward encouraging the authors to work on the manuscript further for the next submission cycle, as I feel like the paper has the *potential* to be excellent after thoughtful revision with a clearer view of the literature.
> > > > > >
> > > > > > [11] Pinto et al. RegMixup: mixup as a regularizer can surprisingly improve accuracy & out-of-distribution robustness *NeurIPS 2022*
> > > > > >
> > > > > > [12] Feng et al. Towards Better Selective Classification *ICLR 2023*

---

> > > > > > > ### Author Response · Authors · 2025-08-07
> > > > > > > **Response (1/2)**
> > > > > > >
> > > > > > > `Point 1`
> > > > > > >
> > > > > > > **The observation that post-hoc logit adjustments can have a significant but inconsistent effect on ranking error ($\varepsilon_{\text{rank}}$) is precisely the phenomenon our framework is designed to explain.**
> > > > > > >
> > > > > > > The reviewer is correct: methods like temperature scaling *can* work, sometimes substantially [2]. However, the fact that they also often fail [3] reveals their core limitation. These methods are **corrective**, not generative. They can only re-organize the ranking information already present in a model's final logits; they are inherently limited in create new ranking signal where none exists. Their success is therefore fragile and entirely conditional on the upstream model's training and architecture. A "vanilla" model produces logits with poor ranking signal, and no amount of post-hoc adjustment can fix that.
> > > > > > >
> > > > > > > In contrast, the methods we highlight—such as Deep Ensembles (DE) and Self-Adaptive Training (SAT)—are fundamentally more powerful because they are **generative**. They are engineered to produce superior correctness rankings from the start, either by leveraging richer feature representations or aggregating diverse model perspectives. They don't just patch a flawed signal; they create a better one. Again, we also want to highlight our extended investigation into loss prediction (Appendix E) which shows that both DE and SAT are consistenly better at predicting their own loss, providing additional evidence for their improved ranking ability.
> > > > > > >
> > > > > > > Therefore, the literature does not contradict our position but rather provides compelling evidence for it. The observed inconsistency of post-hoc methods is a symptom of their mechanistic limitations. Our framework, by isolating the $\varepsilon_{\text{rank}}$ term, provides the necessary theoretical lens to understand *why* these methods are unreliable and why more integrated approaches are required to robustly close the selective classification gap.
> > > > > > >
> > > > > > > ---
> > > > > > >
> > > > > > > `Point 2`
> > > > > > >
> > > > > > > We want to stress that the checklist must be understood within a specific context and with certain caveats:
> > > > > > >
> > > > > > > **Scope is Limited to Selective Classification (SC):** The recommendations in the checklist are tailored **specifically for selective classification**. We cannot expand the scope to include recommendations for related but distinct tasks like Out-of-Distribution (OOD) detection, as the optimal strategies for each can differ significantly. A technique that benefits SC may degrade OOD performance, and vice-versa. The checklist's purpose is to provide actionable advice for a single, well-defined problem, not to serve as a universal guide for all uncertainty-aware machine learning.
> > > > > > >
> > > > > > > **Joint Interactions Remain an Open Question:** The reviewer is correct that many of these recommendations have not been investigated jointly. We are not claiming to have a final, unified theory of error reduction in SC. In fact, after our latest response to reviewer `9vHG`, we **explicitly state that most prior work tackles one error component in isolation**. The complex, joint interactions between these different recommendations remain a critical and open research direction. Our work aims to lay the groundwork for such investigations, and we hope the community continues to produce research that explicitly studies these interactions.
> > > > > > >
> > > > > > > **This is a Checklist, Not a Survey:** It is essential to recognize that the goal of this appendix is **not to provide a comprehensive, encyclopedic treatment of the field**. That is a task better left to a dedicated survey paper. This checklist is intended as a practical, actionable guide based on the best available evidence for improving SC systems *today*. It distills key findings into a usable format for practitioners, but it does not—and cannot—substitute for a full literature review.
> > > > > > >
> > > > > > > **The Checklist is a Dynamic Document:** Finally, this checklist should be viewed as a **living document, not a static set of rules**. It represents our synthesis of recommendations that are currently available and supported by evidence. As the research community produces new work and uncovers more about the intricate interactions between these components, the checklist will inevitably be revised, updated, and refined. It is a starting point, not the final word.
> > > > > > >
> > > > > > > *(continued in next reply)*

---

> > > > > > > > ### Author Response · Authors · 2025-08-07
> > > > > > > > **Response (2/2)**
> > > > > > > >
> > > > > > > > `Point 3`
> > > > > > > >
> > > > > > > > **We respectfully disagree with the reviewer that [12] provides evidence that SAT improved $\varepsilon_{\text{approx}}$ instead of $\varepsilon_{\text{rank}}$.** In our opinion, the cited work by **Feng et al. lacks the formal tools to disentangle improvements coming from better model generalization versus those from better correctness ranking**. Their analysis conflates these two distinct effects, but our framework provides a thought framework to separate them.
> > > > > > > >
> > > > > > > > The core of our argument is this: Any improvement in selective classification performance comes from a reduction in the "selective classification gap". Our paper is the first to formally decompose this gap into its constituent parts, including model approximation error ($\varepsilon_{\text{approx}}$) and ranking error ($\varepsilon_{\text{rank}}$).
> > > > > > > >
> > > > > > > > The work in [12] demonstrates that methods like SAT produce more generalizable classifiers and that a simple MSP is then a better selection mechanism. But it provides no way to determine how much of the total performance gain comes from a better base model (a reduction in $\varepsilon_{\text{approx}}$) versus a more effective ranking of inputs (a reduction in $\varepsilon_{\text{rank}}$). Their methodology simply cannot distinguish between these two separate sources of improvement. **The fact that the rejection head is not used to make rejection decisions does not mean that all improvements instilled into the model during optimization necessarily come from improved approximation. Noticeably, their work uses the performance metric of accuracy at a given coverage which favors models that are more accurate overall (at full coverage) [D from reviewer `9vHG`'s references], further complicating the disentanglement.**
> > > > > > > >
> > > > > > > > Far from undermining our claims, their findings highlight precisely why our decomposition is so crucial. Without it, one is left attributing performance gains to a vague notion of "better generalization," as in [12]. With our framework, we can formally dissect the source of the improvements. Moreover, our loss predition results in Appendix E show that SAT is way better suited at the loss prediction task than MSP/SR.
> > > > > > > >
> > > > > > > > ---
> > > > > > > >
> > > > > > > > **Again, we thank the reviewer for their active participation in the discussion and for their great attention to detail. We are grateful for their expertise and our work has significantly improved thanks to their feedback! While we still hope that we have convinced the reviewer of the holistic value of our work (gap decomposition, role of ranking, connection to loss prediction, controlled experiments, etc.), we are skeptical that further engagement would provide meaningful additional value from this point onwards.**

---

> > > > > > > > > ### Comment · Reviewer_7Yt4 · 2025-08-07
> > > > > > > > >
> > > > > > > > > I appreciate the authors' response and pushback to my previous comment. I think perhaps some of the disagreement stems from imperfect communication on my part. I will take the time to carefully consider our discussion, as well as the discussions with other reviewers before making my final recommendation. There has been a lot of back and forth and I will admit that I was beginning to lose track a little.

---

> ### Comment · Reviewer_7Yt4 · 2025-08-08
>
> I will first briefly respond to the authors' previous comment. I will then follow up in a later comment a summary of my thoughts on the paper.
>
> 1. I agree with the point the authors make about the fundamental potential impact of "corrective" vs "generative" approaches to the problem -- what I was more concerned about was the diminishment of the practical relevance of post-hoc approaches given their demonstrated usefulness in the literature in certain cases. I would be satisfied if the authors re-adjusted their wording to more clearly highlight how and when post-hoc approaches can be useful.
> 2. I was primarily concerned with firm recommendations in the checklist without sufficient clarification/verification on SC-based effectiveness. I had missed the authors' reply to reviewer 9vHG about tagging bullets (as well as other improvements to the checklist), which I believe alleviates this issue sufficiently.
> 3. I agree with the authors' that [12] do not formally disentangle the $\varepsilon$ terms presented in this work, and admit my use such terms in the previous comment was imprecise. However, the results in [2] still question whether SAT leads to improvements solely due to better ranking (as was the narrative presented in the original SAT paper), which call into question the placement of SAT in the checklist. With regards to Appendix E, although I unfortunately do not have the bandwidth to read the mathematics in detail, I am sceptical of the utility of "Loss Prediction advantage", since it is perfectly possible for a model with terribly calibrated probability scores to have zero $\varepsilon_\text{rank}$ (e.g. $p=0.51$ for all correct, $p=0.49$ for all errors). Besides, it seems like the definition of $h(x)$ is unclear in this section between outputing probability scores or hard 1-hot predictions.

---

> > ### Comment · Reviewer_7Yt4 · 2025-08-08
> > **Final thoughts**
> >
> > After rereading my discussion with the authors as well as their discussion with other reviewers, I would like to summarise my thoughts on the submission.
> >
> > - My views on the strengths of the paper remain unchanged from my initial review.
> > - The authors (for the most part) have "patched" the various issues mentioned by myself and other reviewers, as well as taken in feedback relating to paper presentation/communication. I am appreciative of their efforts as the discussion has been extensive.
> >
> > As such, **I do not have a strong reason to reject the submission**; however, due to the *extensive changes* undergone during rebuttal, the inability for reviewers to clearly view the current (post-rebuttal) state of the submitted manuscript, and the issues that exist in the original manuscript if viewed as is, I **strongly believe the paper would benefit from an additional round of peer review** (in agreement with reviewer 9vHG). This would allow the manuscript to be properly evaluated in its current, evolved state, and potentially give it the chance to receive a higher recommendation (I reiterate my positive view on the *potential* of the paper).
> >
> > As such I will **raise my score to borderline accept**, however, I **urge the AC to consider the above when making their final recommendation**.
> >
> > Finally a re-iterate my remaining "niggle" about the use of loss prediction to approximate $\varepsilon_\text{rank}$. It is perfectly possible for a model with terribly calibrated probability scores to have zero $\varepsilon_\text{rank}$ (e.g. $p=0.51$ for all correct, $p=0.49$ for all errors). Since there is only an *upper* bound on the error and no lower bound, comparisons based on the value of the bound are not necessarily valid -- comparing the worst case scenario doesn't necessarily reflect the actual scenario. This is a *minor issue* as it affects ancillary results in the appendix.

---

### Official Review · Reviewer_9vHG · 2025-06-29

**Clarity:** 3
**Significance:** 2
**Originality:** 2
**Rating:** 4
**Confidence:** 3

**Summary:**

They formulate the performance difference of a selective classifier to its perfect ordering scoring (optimality) measured with the *AUACC* (Area Under the Accuracy–Coverage Curve) as the *coverage-uniform selective-classification gap*. Further they differentiate the finite-sample decomposition of the gap dividing it into five parts: 1)“Bayes noise/irreducible”, 2)“appoximation error/capacity”, 3)“ranking error/ranking”, 4)“statistical noise/data” and 5)“implementation or shift-induced slack/optimization & shift”.

They provide a theoretical analysis of closing the gap and show that only calibration that modifies the ranking improves selective classification, with a specific focus on temperature scaling and the reason for its limited effect on the AUACC.

Finally they support their theoretical analysis by means of empirical experiments on: the Toy Two Moons dataset, CIFAR10 & CIFAR100, Stanford Cars and Camelyon 17-WILDS.

**Questions:**

For questions, I refer to the Weaknesses, where I already detailed ways to improve the quality of the work. In general, I am positive about the value of this work, but see it more as introducing a thought framework and guiding beginners in the field, which could be greatly improved by aggregating findings of previous work, as I see the overall novelty and scale of their experiments as limited.

**Ethical Concerns:**

["NO or VERY MINOR ethics concerns only"]

**Final Justification:**

The authors made a great rebuttal; therefore, I increased my score. Several issues with the work crystallized during the rebuttal, which were addressed by the authors. While doing that, however, potentially novel issues became clear due to the multitude of changes. Especially, their main derivation (eq.27) is skipping some important steps and was not adequately described in the submitted version.

Despite my leaning towards acceptance, the multitude of changes makes it hard for me to grasp the manuscript in its updated form. Therefore, I believe another round of reviewing might be beneficial.

**Limitations:**

yes, all limitations except the ones mentioned in the Weaknesses section are addressed.

**Quality:**

3

**Strengths And Weaknesses:**

### Strengths
- The paper provides a clear theoretical thought framework for silent failure cases in selective classification, which the selective classifier does not catch due to the five aforementioned sources of errors.
- The paper is well structured and beautiful if I may say so. Especially, the takeaway boxes substantially improve parsing of the text.
- The appendix is well organized and, in combination with the code provided in the supplementary material, which is also well structured, seems to provide all the information necessary to replicate their empirical results.
- All of this combined can make it a nice starting point for people working themselves into the field of selective classification and on which areas to focus to improve selective classifiers, especially in combination with Appendix C, the “Practitioner Checklist for Tightening the Selective-Classification Gap”.

## Weaknesses
**Formulation**:
- Given the stated purpose in (lines 35-36): “For my finite model on finite data, what aspects of the learning setup will actually move my trade-off curve closer to the upper bound?”
- The algebraic decomposition of the selective classification gap (here a) in equation 27 boils down to an inequality of the form $a <= a + b$ without a clear reasoning why we should care about $b$ if the goal is to improve $a$ as the upper bound appears to be defined as the perfect ranking upper bound, which is defined as $a$.
    - Please clarify what the exact goal of the finite-sample decomposition is in a simple manner, and especially explain the reason for using this decomposition.
- As the formulation of the selective classification gap seems to be very similar –if not identical– to that of the e-AURC [D], this should be discussed.
- Overall, the theoretical formulations could be greatly improved by clarifying the relations to existing literature and metrics.


**Related Works**:
- Currently the results of this paper are not discussing some recent works which have been highly prominent in the community like [A] (ICLR Oral), [B] (NeurIPS Spotlight) and [C] (NeurIPS). This substantially reduces the quality of the text and reduces its value as starting point for people to familiarize themselves with the topic of selective classification.
    - Specifically [B] introduces the AUGRC which should be discussed in Section 2 paragraph “Accuracy-coverage tradeoff evaluation.” as it essentially reweights the AURC with the coverage which reduces the impact of the very low coverage area.
    - The finding of [A] based on a large-scale empirical study reveals that “None of the prevalent methods proposed in the literature is able to outperform a softmax baseline across a range of realistic failure sources,” which would help motivate the selection of MSP in Section 4.
        - Further, the reasoning in [A] goes against the reasoning here because the goal is to reduce the selective classification gap instead of the best overall classifier. This is worth discussing what your view means for the practitioner.
    - A detailed discussion of the difference to the e-AURC [D] is missing where it is analyzed based on formulas.
- See last point in “Formulation”.


**Novelty**:
- Playing devils advocate here: That label noise (Bayes), overall model performance (capacity), sample size (data) and optimization errors or distribution shifts (optimization & shifts) influence the overall model performance and that uncertainty estimation (ranking) influences the selection process is hardly new to anyone in the field.
    - I see the main value in the thought framework and handing solutions. Based on this I believe that more citations to the work referenced as solutions in App. C would greatly increase the value of this work as well as the works which indicate that this is a reliable solution for the use case of selective classification to make it more valuable for future reference.
- Further the novelty that methods which induce changes in the ranking lead to substantial improvements is rather limited as this is true for basically any metric which is based on ordering of scores (e.g. AUROC, AUPRC…)


**Evaluation**:
- I do not agree with the authors that they perform a “large scale” evaluation. The evaluation works like [A] and [B] spans more methods and more datasets while [C] uses more methods and ImageNet instead of the small scale CIFAR, Stanford and Camelyon datasets.
    - Solutions: (1) rephrase this, (2) In an optimal scenario I encourage the authors to reinterpret the results from [A], [B] and [C] in their framework.
- General weaknesses with regard to the evaluation: They only evaluate four different selective classification methods 1) MSP, 2) Temp. Scaled Softmax, 3) Self-Adaptive Training, 4) Deep Ensemble. All of which are simple to implement and 1 & 2 are essentially identical with regard to the overall setup. Further many of the Practical Guidelines in App. C are not explicitly shown (e.g. how RandAugment improves the model [see point above for a solution not necessitating extension of the evaluation]).


**Minor points**:
- In Sections 4.1 and 4.2 in Setup: there are too many enumerations with (i) to (iii) I would appreciate to do (i)...(iii) and (a)... (z) to make the distinction across them clearer.



### Citations
[A] Jaeger, P. F., Lüth, C. T., Klein, L., & Bungert, T. J. (2023). A call to reflect on evaluation practices for failure detection in image classification. The Eleventh International Conference on Learning Representations. https://openreview.net/forum?id=YnkGMIh0gvX

[B] Traub, J., Bungert, T. J., Lüth, C. T., Baumgartner, M., Maier-Hein, K., Maier-hein, L., & Jaeger, P. F. (n.d.). Overcoming common flaws in the evaluation of selective classification systems. The Thirty-Eighth Annual Conference on Neural Information Processing Systems. https://openreview.net/forum?id=2TktDpGqNM&noteId=MJx9cPynNy

[C] Mucsányi, B., Kirchhof, M., & Oh, S. J. (2024). Benchmarking uncertainty disentanglement: Specialized uncertainties for specialized tasks. In A. Globerson, L. Mackey, D. Belgrave, A. Fan, U. Paquet, J. Tomczak, & C. Zhang (Eds.), Advances in neural information processing systems (Vol. 37, pp. 50972–51038). Curran Associates, Inc. https://proceedings.neurips.cc/paper_files/paper/2024/file/5afa9cb1e917b898ad418216dc726fbd-Paper-Datasets_and_Benchmarks_Track.pdf

[D] Yonatan Geifman, Guy Uziel, and Ran El-Yaniv. Bias-reduced uncertainty estimation for deep neural classifiers. arXiv preprint arXiv:1805.08206, 2018

---

> ### Author Rebuttal · Authors · 2025-07-29
>
> We sincerely thank the reviewer for their in-depth feedback. We are especially grateful for their recognition of the paper’s **structured presentation, clear theoretical framing,**  and its potential as a **starting point for newcomers to SC**. At the same time, we acknowledge the reviewer’s **concerns regarding the novelty, formulation, and scope of evaluation**. We have taken these suggestions seriously and provide clarifications, additional related work discussion, and rewordings throughout our rebuttal to address them in detail.
>
> ---
>
> ``1.1) Purpose of lines 35-36.``
>
> We would appreciate additional clarification, as we are unsure what the reviewer is specifically asking.
>
> ---
>
> ``1.2) Algebraic decomposition of Equation 27.``
>
> We apologize for the confusion—Eq. 27 is *not* intended as the tautological inequality $a \le a + b$. Rather, **by adding and subtracting $\mathbb{E}[\eta_h \mid A_c^{\star}]$, we obtain an exact identity**
>
> $$
> \Delta(c) = \varepsilon_{\text{rank}}(c) + \varepsilon_{\text{approx}}(c) + \varepsilon_{\text{Bayes}}(c),
> $$
>
> where each non-negative term isolates a distinct failure mode:
> **(i)** $\varepsilon_{\text{Bayes}}$ is the irreducible label noise among the accepted points;
> **(ii)** $\varepsilon_{\text{approx}}$ captures errors that persist even if ranking were perfect (e.g., due to limited model capacity or training); and
> **(iii)** $\varepsilon_{\text{rank}}$ quantifies the loss due solely to mis-ordering examples.
>
> **This factorization is therefore diagnostic, not merely algebraic**: by measuring the three components in Sec. 4, we show, for instance, that ranking error dominates on CIFAR-100, pointing practitioners toward improving uncertainty estimation rather than enlarging the model. We have added a concise explanation of this objective immediately before Eq. 27 to prevent further misunderstanding.
>
> ---
>
> ``1.3) Relationship to e-AURC [D].``
>
> The reviewer is correct that our formulation of the SC gap shares similarities with the definition of the Excess-AURC (E-AURC) introduced in [D], as both quantify deviations from an ideal abstention policy. However, **the two differ in both granularity and perspective**. Our gap measures pointwise accuracy shortfall at a fixed coverage level relative to a perfect oracle with the same base accuracy, whereas E-AURC aggregates excess risk across all rejection levels relative to an optimal ranking function. To clarify this connection, we have explicitly acknowledged the similarity between our gap and E-AURC directly following Definition 4 in the updated draft. We would also be happy to include a more detailed comparison of the two performance metrics in the appendix.
>
> ---
>
> ``2.1) Discussion of recent works [A,B,C].``
>
> We thank the reviewer for pointing us to these influential recent works. **We have incorporated their insights in the revised manuscript** as follows:
>
> In §2 “Accuracy–coverage trade-off evaluation” we added:
>
> > Recently, [B] introduced the Area Under the Goals-Reweighted Curve (AUGRC), which multiplies accuracy by coverage before integration, thereby down-weighting the very-low-coverage tail that can dominate the traditional AURC.
>
> > Complementary to metric design, [C] provides a comprehensive benchmark that disentangles sources of uncertainty, underscoring the need for task-aware evaluation of selective classifiers.
>
> In §4.2 “Setup” we added:
>
> > Our inclusion on the maximum-softmax-probability baseline is motivated by the large-scale study of [A], who find that MSP proves to be a hard-to-beat baseline.
>
> ---
>
> ``2.2) [A] goes against our reasoning.``
>
> Certainly—here is a more concise version that integrates your collaborator’s feedback while keeping the overall length close to the original:
>
> ---
>
> First, we note that **\[A] did not evaluate Self-Adaptive Training (SAT) or Deep Ensembles (DE)**. Instantiated to support our theoretical formulation, our empirical results in §4.2 and Appendix E.5 show that **SAT and DE consistently narrow the selective-classification gap better than MSP**—by improving ranking quality—and are also more reliable at predicting their own loss. Second, we agree that practitioners ultimately care about the best possible performance at the chosen coverage. In that sense, **minimizing the selective-classification gap complements accuracy maximization**: it reveals whether remaining error stems from noise, ranking, or capacity limits, and thus which lever to pull (e.g., stronger backbone vs. better uncertainty scoring). This disentanglement also clarifies whether improvements from compute-heavy methods like DE come from gap reductions or generic accuracy boosts—helping practitioners identify more efficient alternatives. Please let us know if we have misinterpreted your concern.
>
> ---
>
> ``3.1) Role of isolated gap quantities is hardly new to anyone in the field.``
>
> We agree that practitioners often recognize each factor qualitatively. However, to our knowledge, no prior work **(i)** formalizes *all five* influences in a single finite-sample, coverage-uniform inequality, **(ii)** proves that every SC gap can be *exactly upper-bounded* by these terms, and **(iii)** demonstrates—via controlled experiments—that each term can be *independently* measured and targeted. Our decomposition thus **turns familiar intuitions into a quantitative error budget that guides design**: it shows why monotone post-hoc calibration cannot improve ranking, when added capacity yields diminishing returns due to Bayes noise, and how shift-induced slack can limit SC gains. If the reviewer knows of prior work with this combination of decomposition, bounds, and empirical disentanglement, we would be grateful for the reference and acknowledge it. See extended discussion for Reviewer VZwS.
>
> ---
>
> ``3.2) More citations in App. C and in the main body.``
>
> This is a good suggestion; we agree with the reviewer that the **checklist would benefit from additional citations** to prior work that has proposed or validated the listed solutions. In the revised version, we will enrich Appendix C by explicitly citing the relevant methods alongside each entry, including both original sources and follow-up studies that confirm their utility in selective classification. At the same time, **we will also include relevant citations in the main body**. We hope that this will help readers trace the origins of each recommendation and better understand the contexts in which they have been shown to be effective.
>
> ---
>
> ``3.3) Ranking insight and connection to AUROC, AUPRC.``
>
> We appreciate the reviewer’s observation that metrics like AUROC/AUPRC improve with better score ordering. **Our goal is not to reiterate this general fact, but to make it operational in the context of selective classification**. Specifically, we isolate the ranking error term and show that **(i)** monotone calibration methods cannot reduce it, while **(ii)** reranking techniques can. Unlike threshold-averaged metrics, **SC directly evaluates accuracy at fixed coverage levels, where a poor ranking can have a more immediate cost**. By embedding ranking error alongside Bayes noise, approximation error, and slack in a single inequality—and validating each term through controlled experiments—we provide practitioners with a diagnostic toolkit to identify what limits performance and how to improve it.
>
> ---
>
> ``4.1) “Large scale” evaluation appears to be overclaiming.``
>
> Our paper devotes substantial space to **(i)** developing a theoretical decomposition of the selective-classification gap, **(ii)** identifying which calibration mechanisms can or cannot tighten it, and **(iii)** connecting selective prediction to loss prediction. **The experiments are designed as controlled diagnostics that vary one interpretable factor at a time to isolate its effect on each bound term**. That said, we agree that the phrase “large-scale evaluation” overstates the current scope. We have revised the paper to the more accurate wording **“systematic, controlled evaluation on synthetic and vision benchmarks.”** We also note our response to Reviewer VZwS, who requested **additional LLM results**.
>
> ---
>
> ``4.2) Only four different SC methods are implemented: 1) MSP 2) TEMP 3) SAT 4) DE.``
>
> We chose these four methods not for sheer breadth but as **controlled, representative instantiations of the key scoring strategies** in our decomposition: MSP/TEMP to illustrate that monotone recalibration only affects approximation error, SAT as a feature‐aware reranker, and DE as an ensemble‐based reranker. This ablation‐style setup lets us cleanly attribute gap reductions to each mechanism. Although the implementations are simple, these methods are widely used because of their ease of implementation and the insights into how each method maps to our theoretical error terms are novel. We will clarify this in §4.2 and, if desired, include a brief appendix study of additional scoring approaches.
>
> ---
>
> ``4.3) Many recommendations from App. C are not explicitly experimented on.``
>
> Indeed, the recommendations in Appendix C consist of a combination of insights from our work as well as best practices in the SC literature. In the revised Appendix C we now **distinctly label each recommendation** as either **(i)** an original insight from our theoretical insights and controlled experiments; or **(ii)** a best practice drawn from the SC literature. As previously mentioned, we have added targeted citations so that readers can immediately see the evidence supporting each guideline.
>
> ---
>
> ``5) Enumerations in 4.1 & 4.2.``
>
> We have fixed this as per the reviewer's suggestion.
>
> ---
>
> **We hope that our rebuttal has addressed the reviewer’s questions and concerns and that the reviewer considers raising their individual and overall score(s). Our work has already significantly improved thanks to their feedback. We are happy to further engage with them during the discussion period!**

---

> ### Comment · Reviewer_9vHG · 2025-08-04
>
> I thank the authors for their detailed rebuttal.
>
> While I share Reviewer 7Yt4’s view on the potential value of this work and would like to raise my score, I remain unconvinced about the additional contribution over the E-AURC. This is primarily due to the continued lack of clarity regarding the mathematical decomposition of the gap (which I have now outlined more explicitly) as well as the strong similarity to E-AURC, which has not been sufficiently addressed.
>
>
> ---
>
> **Points 1.1 and 1.2 should be seen combined. I am sorry if this led to confusion.**
>
> As the stated purpose is to explicitly reduce the SC gap, it is important to assess how the proposed decomposition helps to do this. As of now, I am unsure how it does so.
>
> Based on the definitions in the manuscript (see 23 for $\varepsilon_{\text{rank}}(c)$, below 27 and Point 1 for $\mathbb{E}[\eta_h \mid A_c^\star]=\overline{acc}(a_{\text{full}}, c)$, below equation 26 step 1 for $acc_{c}(h, g)=\mathbb{E}[\eta_h \mid A_c]$) the following statement holds true:
> $$
> \Delta(c) := \overline{acc}(a_{\text{full}}, c) - acc_{c}(h, g) = \mathbb{E}[\eta_h \mid A_c^\star] - \mathbb{E}[\eta_h \mid A_c] = \varepsilon_\text{rank}(c)
> $$
>
> Therefore, based on this, the authors prove that (see answer 1.2 and proof in Theorem 1):
> $$
> \Delta(c) = \varepsilon_\text{rank}(c) = \varepsilon_{\text{rank}}(c) + \varepsilon_{\text{approx}}(c) + \varepsilon_{\text{Bayes}}(c)
> $$
> Therefore $\varepsilon_{\text{approx}}(c) + \varepsilon_{\text{Bayes}}(c)=0$ where each of these must satsify $\geq 0$.
> If this is not equal but greater than or equals, then it boils down to:
> $$
> \Delta(c) = \varepsilon_\text{rank}(c) \leq \varepsilon_{\text{rank}}(c) + \varepsilon_{\text{approx}}(c) + \varepsilon_{\text{Bayes}}(c)
> $$
> But the question is then: how does reducing the other error sources influence the ranking error based on this decomposition?
>
>
> Based on this, it appears that the gap only measures the difference from the perfect ranking.
>
> ---
>
> **Point 1.3 -- Please analyze the similarity of the SC gap to the E-AURC based on Equations instead of a text discussion**
>
> I asked for an analysis of the difference between the proposed SC gap and the E-AURC, especially in the form of a mathematical formulation. Please do this here, and also add it to the paper.
>
> To my understanding, the main argument is that the SC gap operates based on a single coverage point, whereas the E-AURC is only defined as an integral over the entire curve. However, the work introducing the E-AURC already shows that the excess risk can be computed for each coverage point (see [Geifman et al. 2019](https://arxiv.org/pdf/1805.08206) Figure 1).
>
>
> Further, in the manuscript in Table 1, the measurement is called GAP when according to lines 227-229 the integral is taken (in the new results table it is also called Gap Area) which would makes the metric actually used for numerical comparisons identical to the E-AURC, if I am not mistaken based on the point above (1.1-2).
>
> I would highly encourage renaming the measurement to the E-AURC in this case to give proper attribution to the people who originally proposed this measure.
>
> It could be that this stems from my lack of knowledge. However, as the authors are definitely aware of this work (as they cite it in their manuscript) and are also aware of the similarity it is important that the differentiating factor between these two metrics is as clearly defined as possible.
>
> This is paramount to ensure that future works can properly reference both works for their contributions.
>
> ---
>
> **4.3 I agree with Reviewer 7Yt4, please provide explicit examples, which advice is given alongside the citations.**

---

> > ### Author Response · Authors · 2025-08-06
> > **Response (1/2)**
> >
> > We appreciate the reviewer’s active involvement in the discussion phase and their attention to detail. We also apologize for our delay in responding to the reviewer.
> >
> > ---
> >
> > ``Points 1.1 + 1.2``
> >
> > **We respectfully but firmly disagree with the reviewer’s suggestion that our decomposition reduces to a single ranking error term $\varepsilon_\text{rank}(c)$. This conclusion rests on a flawed premise that conflates two different oracle quantities.** The reviewer's argument hinges on the equality $\overline{\text{acc}}(a_\text{full}​,c)=\mathbb{E}[\eta_h​∣A_{c}^*​]$. This is incorrect. As we define them:
> >
> > - $\overline{\text{acc}}(a_\text{full}​,c)$ is the **absolute theoretical bound** for any classifier with full-coverage accuracy $a_\text{full}$​. It represents a idealized accuracy-coverage curve.
> > - $\mathbb{E}[\eta_h​∣A_{c}^*​]$ is the **model-specific oracle accuracy**. It is the best possible selective accuracy for our specific model $h$, achieved with a perfect ranking of examples by its own correctness posterior $\eta_h​(x)$.
> >
> > By definition, a model-specific oracle can, at best, match the absolute theoretical bound. Therefore, the correct relationship is the inequality: $\overline{\text{acc}}(a_\text{full}​,c)\geq\mathbb{E}[\eta_h​∣A_{c}^*​]$.
> >
> > This inequality is strict for any non-ideal model whose correctness probabilities are not perfectly stratified. The gap between these two terms is fundamental to understanding selective classification performance and is precisely what the other two components of our decomposition capture. So while the reviewer is correct that $\Delta(c)$ contains the ranking error, it contains more than just ranking. An alternate derivation compared to the paper that maintains our original three-component structure is as follows:
> >
> > $$
> > \Delta(c):= \overline{\text{acc}}(a\_{\text{full}}, c) - \text{acc}\_c(h,g) \ = \overline{\text{acc}}(a\_{\text{full}}, c) - \mathbb{E}[\eta\_h | A_c] \quad (\text{by definition of } \text{acc}\_c)
> > $$
> >
> > By adding and subtracting the model-specific oracle accuracy, $\mathbb{E}[\eta_h​∣A_{c}^*​]$ we isolate the ranking error:
> >
> > $$
> > \Delta(c) = ( \overline{\text{acc}}(a\_{\text{full}}, c) - \mathbb{E}[\eta\_h | A_c^{\star}] ) + ( \mathbb{E}[\eta\_h | A_c^{\star}] - \mathbb{E}[\eta\_h | A_c] ) \ = \underbrace{( \overline{\text{acc}}(a\_{\text{full}}, c) - \mathbb{E}[\eta\_h | A_c^{\star}])}\_{\text{Model Quality Gap}} + \underbrace{\varepsilon\_{\text{rank}}(c)}\_{\text{Ranking Error}}
> > $$
> >
> > **The reviewer's analysis incorrectly assumes the "Model Quality Gap" is zero.** This gap is not only non-zero but is essential. It quantifies how far our model $h$ is from the absolute ideal, even when its own examples are ranked perfectly. This sub-optimality is attributable to two sources:
> > - **Approximation Error ($\varepsilon_{\text{approx}}$)**: The deviation of our model's predictions from the Bayes-optimal classifier.
> > - **Bayes Error ($\varepsilon_{\text{Bayes}}$)**: The irreducible error from inherent data ambiguity.
> >
> > Our theorem correctly shows that this Model Quality Gap is bounded by the sum of $\varepsilon_{\text{approx}}$ and $\varepsilon_{\text{Bayes}}$. Therefore, the full gap is bounded by the sum of all three components, as stated in our paper:
> >
> > $$
> > \Delta(c)≤\varepsilon_{\text{Bayes}}(c) + \varepsilon_{\text{approx}}(c) + \varepsilon_{\text{rank}}(c)
> > $$
> >
> > This directly answers the reviewer's final question. The gap measures far more than ranking performance. Reducing approximation error (e.g., with a better model) or having less Bayes error (cleaner data) closes the "Model Quality Gap," thereby reducing $\Delta(c)$ independently of the ranking error. Our three-part decomposition is not only intact but necessary to account for all distinct sources of error.
> >
> > *(continued in next reply)*

---

> > > ### Author Response · Authors · 2025-08-06
> > > **Response (2/2)**
> > >
> > > ``Point 1.3``.
> > >
> > > We apologize for the brevity in our original response. Unfortunately, we had to fit our full rebuttal to all the reviewer’s points within 10k characters. We elaborate on the differences and similarities between our SC gap and the E-AURC below. **The reviewer is correct that the two concepts are intimately related, and we are happy to provide a formal analysis to clarify their distinction and connection.** Below, we provide a mathematical formulation and then outline our proposed revisions to the manuscript.
> > >
> > > **Mathematical analysis: SC Gap vs. E-AURC**: We formally define both metrics to analyze their relationship.
> > >
> > > 1. **Our SC Gap $\Delta (c)$:** As defined in our manuscript (Definition 2), the SC gap at a specific coverage level $c$ is the difference between the accuracy of a perfect-ordering oracle, $\overline{\text{acc}}(a\_{\text{full}}, c)$, and the accuracy of the selective classifier $(h, g)$ at that coverage, $\text{acc}\_c(h, g)$.
> > > $$
> > > \Delta(c) = \overline{\text{acc}}(a\_{\text{full}}, c) - \text{acc}\_c(h, g)
> > > $$
> > > This metric quantifies the performance drop from the ideal case at a *single, specific point* on the coverage spectrum.
> > >
> > > 2. **Geifman et al.'s E-AURC:** The E-AURC is defined as the difference between the Area Under the Risk-Coverage Curve (AURC) for a given confidence function $\kappa$ and the AURC for an optimal, in-hindsight confidence function $\kappa*$:
> > > $$
> > > \text{E-AURC} = \text{AURC}(\kappa, f) - \text{AURC}(\kappa*, f)
> > > $$
> > > The $\text{AURC}$ itself is the integral of the selective risk, $R(c)$, over the coverage $c$ from 0 to 1 (in the continuous case): $\text{AURC}(\kappa, f) = \int_0^1 R(c; \kappa, f) dc$ where $R(c; \kappa, f)$ is the selective risk at coverage $c$ for the classifier $f$ using confidence function $\kappa$.
> > >
> > > **The connection:** The selective accuracy we use, $\text{acc}\_c$, is simply $1 - R(c)$, where $R(c)$ is the selective risk (assuming 0/1 loss). Therefore, our SC Gap can be rewritten in terms of risk:
> > >
> > > $$
> > > \Delta(c) = (1 - R\_{\text{oracle}}(c)) - (1 - R\_{\text{actual}}(c)) = R\_{\text{actual}}(c) - R\_{\text{oracle}}(c)
> > > $$
> > >
> > > This shows that our $\Delta(c)$ is precisely the **excess selective risk** at coverage $c$. The E-AURC from Geifman et al. is the integral of this pointwise excess risk over all coverage levels:
> > >
> > > $$
> > > \text{E-AURC} = \int\_0^1 R_{\text{actual}}(c) dc - \int\_0^1 R_{\text{oracle}}(c) dc = \int_0^1 \big(R\_{\text{actual}}(c) - R\_{\text{oracle}}(c)\big) dc = \int\_0^1 \Delta(c) dc
> > > $$
> > >
> > > **TL;DR:** The reviewer's understanding is correct. Our pointwise \$\Delta(c)\$ is the integrand of the E-AURC. The "Gap Area" or integrated gap we used in our original empirical evaluation is mathematically equivalent to the E-AURC.
> > >
> > > Our primary contribution in Section 3 is a formal *decomposition* of this gap at a *fixed coverage level* $c$. For this analysis, a pointwise metric like $\Delta(c)$ is necessary and more direct than working with the integrated E-AURC. However, we agree that the connection should be made explicit and proper attribution given, especially in the empirical section where we use the integrated metric.
> > >
> > > **Proposed changes to the paper**: Based on this analysis, we propose the following compromise which we believe addresses the reviewer's concerns:
> > >
> > > 1. *We have updated Definition 4 to explicitly acknowledge its relationship to the E-AURC.*
> > > > **Definition 2 (Selective Classification Gap).**
> > > >
> > > > *(start same as in paper)*
> > > >
> > > > This gap $\Delta(c)$ represents the *excess selective risk* at a given coverage $c$. Its integral over the entire coverage spectrum, $\int_0^1 \Delta(c) dc$, is equivalent to the **Excess-AURC (E-AURC)** metric proposed by Geifman et al., 2019.
> > >
> > > 2. *Our theoretical derivation in Section 3.3 will continue to use $\Delta(c)$* because our goal is to provide a formal decomposition of the gap at a specific coverage level, for which the pointwise notation is most natural.
> > >
> > > 3. *In the empirical section (including Table 1), we have renamed our integrated gap metric to "E-AURC"* to give proper attribution and use the established terminology for this aggregate measure. We have added a sentence explicitly stating that for our numerical comparisons, we compute the E-AURC by integrating our empirical gap, $\widehat{\Delta}(c)$, across all coverage levels. For instance, lines 227-228 now read as follows:
> > > > For each setting, we compute the Excess-AURC (E-AURC) (Geifman et al., 2019) by integrating the empirical gap $\widehat{\Delta}(c)$ across all coverage levels.”
> > >
> > > We believe these changes will make the relationship between the metrics crystal clear, give due credit for the E-AURC, and properly situate our novel contribution, which is the *decomposition* of the pointwise gap.
> > >
> > > ---
> > >
> > > ``Point 4.3``
> > >
> > > **Please see our response to reviewer 7Yt4.**
> > >
> > > ---
> > >
> > > **We hope that our response has clarified the reviewer’s concern and continue to be available for further clarification requests.**

---

> ### Comment · Reviewer_9vHG · 2025-08-06
>
> I thank the authors for their detailed response.
>
> ---
>
> I am fully satisfied with their **Mathematical analysis: SC Gap vs. E-AURC** as well as their proposed changes.
>
> ---
>
> Regarding point 4.3.
>
> Thank you for this detailed answer.
>
> I am highly positive about these changes.
>
> However, I would encourage the authors to also explicitly search for works that show that the stated changes improve Selective Classification Performance to give proper attribution to works that already showcased this explicitly.
>
> Further, if you can not find explicit references or it does not directly follow from the experiments in this manuscript, add a differentiation to show that these are based on a theoretical analysis and remain to be shown explicitly.
>
> I will leave it up to the authors to follow up on these changes.
>
> I deem this important to ensure that practitioners reading this chapter can clearly differentiate between hypotheses/theoretical results and best practices, which have already been shown (alongside how, based on the citation) and are gathered in one place.
>
>
>
> ---
>
> **Regarding Point 1.1 and 1.2**
>
> I appreciate the clarification from the reviewers and would encourage them to write this exact reasoning down in equation 27.
>
> However, I wish to clarify that **I do not assume this statement to be true**:
> $$
> \overline{acc}(a_{\text{full}}, c) = \mathbb{E}[\eta_h \mid A_c^\star]
> $$
> **Rather, it is directly written in the manuscript** as I already pointed out in my previous answer (directly below equation 27 in point 1 following "Explanation of the two labelled inequalities." ... "(rank) uses the definition of the oracle: $\overline{acc}(a_{\text{full}}, c) = \mathbb{E}[\eta_h \mid A_c^\star]$").
>
> Am I not reading it correctly? Is this a mistake in the manuscript, or is this something else?

---

> > ### Author Response · Authors · 2025-08-06
> >
> > ``Points 1.1 and 1.2``
> >
> > **The reviewer is correct that step 1 following Eq (27) mistakenly includes the equality** $\overline{\text{acc}}(a\_\text{full}​,c)=\mathbb{E}[\eta\_h​∣A\_{c}^*​]$. We apologize for this confusion and have since proposed a fix to the proof based on our reply to the reviewer. We note that the theorem and the three-part decomposition remain correct.
> >
> > The derivation in Step 1 of the proof should proceed as follows:
> > $$
> > \Delta(c) := \overline{\text{acc}}(a\_{\text{full}}, c) - \text{acc}\_c(h,g) = \overline{\text{acc}}(a\_{\text{full}}, c) - \mathbb{E}[\eta\_h | A\_c]
> > $$
> > We add and subtract the model-specific oracle accuracy, $\mathbb{E}[\eta_h | A_c^{\star}]$:
> > $$
> > \Delta(c) = \underbrace{\left( \overline{\mathrm{acc}}(a\_{\text{full}}, c) - \mathbb{E}[\eta\_h | A_c^{\star}] \right)}\_{\text{Model Quality Gap}} + \underbrace{\left( \mathbb{E}[\eta\_h | A_c^{\star}] - \mathbb{E}[\eta\_h | A\_c] \right)}\_{\varepsilon\_{\text{rank}}(c)}
> > $$
> > The second term is precisely the ranking error, $\varepsilon_{\text{rank}}(c)$. The first term, the "Model Quality Gap," represents how far even a perfectly-ranked version of our model $h$ falls short of the absolute ideal. This gap is bounded by the other two error sources, which are defined on the accepted set $A_c$:
> >
> > $$\overline{\mathrm{acc}}(a\_{\text{full}}, c) - \mathbb{E}[\eta\_h | A\_c^{\star}] \le \varepsilon\_{\text{approx}}(c) + \varepsilon\_{\text{Bayes}}(c)$$
> >
> > Combining these gives the original theorem. We have added this discussion and have also updated the respective paragraph in the proof as follows.
> > > **Explanation of the two labelled inequalities.**
> > >
> > > 1. **(rank)** This step isolates the ranking error, $\varepsilon_{\text{rank}}(c) := \mathbb{E}[\eta_h \mid A_c^{\star}] - \mathbb{E}[\eta_h \mid A_c]$.  The inequality holds because the remaining term from the previous line, $\overline{\operatorname{acc}}(a_{\text{full}}, c) - \mathbb{E}[\eta_h \mid A_c^{\star}]$, is a non-negative quantity that is bounded by the error sources introduced in the next step.
> > >
> > > 2. **(approx+Bayes)** *(as before)*
> >
> > **We sincerely appreciate the reviewers' care in examining our detailed derivation and their attention to detail.**
> >
> > ---
> >
> > ``Point 1.3``
> >
> > We are happy to hear that the reviewer is supportive of this change!
> >
> >  ---
> >
> > ``Point 4.3``
> >
> > We are happy to hear that the reviewer also feels positive about these changes to our checklist. Indeed, we agree that practitioners indeed should be able to see at a glance which checklist items are **evidence-backed** and which remain **theoretical hypotheses**.
> > As such, we further propose the following improvements:
> > - **Visual differentiation.** In Appendix C we now also tag each sub-bullet with either **\[E] Empirical** (demonstrated SC improvement in the cited work) or **\[T] Theoretical** (sound for the targeted error term but not yet validated end-to-end for SC).
> > - **Clarified scope.** Extending on the previous point, we explicitly note that most prior studies tackle one error component in isolation and that joint interactions across recommendations (e.g., how data augmentation and recalibration together affect $\varepsilon\_\text{rank}$) remain an open research direction. We hope that the community continues to produce work that explicitly studies such interactions.
> > - **Living checklist.** A closing paragraph now states that the checklist is versioned and will be updated as the community produces stronger empirical evidence. We are happy to update this checklist over time and also create a corresponding project website which routinely incorporates new insights from the SC community for even easier access and maintenance.
> >
> > ---
> >
> > **Again, we thank the reviewer for their active participation in the discussion. We are grateful for their expertise and our work has significantly improved thanks to their feedback! We hope that the reviewer re-considers their evaluation of our work in light of these changes.**

---

> > > ### Comment · Reviewer_9vHG · 2025-08-06
> > > **Final Rebuttal Answer**
> > >
> > > The authors’ efforts in addressing the concerns are duly noted, and the quality of the manuscript has clearly improved as a result. I will raise my score accordingly.
> > >
> > > That said, given the multitude of changes made in the revision, along with the oversights present in the originally submitted version, I found it challenging to keep track of all modifications while reviewing fully. I therefore believe the paper could benefit from another round of thorough reviewing to ensure clarity and consistency throughout. This would also give the paper the chance to have a greater impact.
> > >
> > > **My main point is as of now:**
> > > The proof resulting in the bound (eq 27) could be shown more clearly and in more steps. As this decomposition is the main contribution of the paper, it would be very helpful to make it as clear as possible to the reader how it is derived.
> > >
> > > Further, it would also be helpful to be clear whether equation 27 is an equality or an inequality, as somehow both were claimed throughout the rebuttal and also in the manuscript.

---

### Official Review · Reviewer_2rvR · 2025-07-03

**Clarity:** 4
**Significance:** 3
**Originality:** 4
**Rating:** 5
**Confidence:** 3

**Summary:**

The paper analyzes the problem of learning selective classifiers, which are classifiers that have the option to abstain on inputs deemed unreliable, reducing the risk of costly errors. Their analysis is based on the gap between the accuracy-coverage tradeoff of the classifier, which quantifies the degradation of performance as the model accepts a broader set of inputs, and a perfect-ordering upper bound, which assumes an oracle that orders the example according to the likelihood of error by the classifier. They show that this gap can be decomposed into five (not necessarily non-overlapping) components that measure different potential sources for this gap: Bayes noise, approximation error, ranking error, statistical noise and implementation of shift-induced slack. Through controlled experiments with both synthetic and real data, they isolate these components, showing the effect of different design choices for selective classifiers that impact the accuracy coverage tradeoff gap. In particular, their results show that calibration methods that are monotonic (i.e. do not affect ranking) do not improve the accuracy coverage tradeoff gap, which was an open question in previous research with contradictory results.

**Questions:**

How is the selective classification gap affected by overfitting? What component does it affect? Have you seen cases in practice where increasing model capacity resulted in a worse accuracy coverage tradeoff?

In the abstract you say that you provide concrete design guidelines for building selective classifiers. Could you list exactly are these concrete guidelines? (I haven’t read the Appendix).

**Ethical Concerns:**

["NO or VERY MINOR ethics concerns only"]

**Final Justification:**

I had originally recommended (4) accept. After reading the other review and rebuttals, I saw that some of the other reviewers had lower scores due mainly to considerations about novelty and clarity. In particular, they requested clarifications about the relationship between the proposed theoretical formulation and previous formulations. While I agree that these were weaknesses in the paper that I had overlooked, I believe that the authors have addressed the concerns satisfactorily in their rebuttal and can improve the final version, so I will keep my score.

**Limitations:**

Yes.

**Paper Formatting Concerns:**

No formatting issues in this paper.

**Quality:**

3

**Strengths And Weaknesses:**

Strengths:
-The paper is well-written, with a good description of the area and recent research, positioning well the contributions of the paper, the conclusions derived from their theoretical and empirical validation and the limitations of the work.
-The experimental analysis both on synthetic and real data is very well designed to show the effect of the different sources of errors
-The paper presents an original analysis of the selective classifier learning problem, introducing new concepts (the selective classification gap) and different ways to understand the impact of different design choices.


Weaknesses:
-Although the formulation is clean and elucidates the problem, it is a little hard to see practical, non-obvious, design guidelines for building selective classifiers coming out of this formulation. Even monotonic calibration could in principle improve approximation error, so we can only find out whether it works through controlled experiments (as done in the paper).
-The paper does not mention overfitting and the effects that this may cause on the selective classification gap — all the experiments show increased performance when the capacity of the model increases in terms of the accuracy-coverage tradeoff gap, but we know that in practice this is not always true because of overfitting.

---

> ### Author Rebuttal · Authors · 2025-07-29
>
> We thank the reviewer for their thoughtful and constructive positive review. We are glad the reviewer finds the **paper is well-written, the analysis original, and the experiments carefully designed**. We appreciate the **concerns raised regarding the practical design takeaways and the role of overfitting**, and we address both points in detail below.
>
> ---
>
> ``1) It is hard to see practical, non-obvious, design guidelines for building selective classifiers coming out of this formulation?``
>
> We want to clarify that our **contribution goes beyond establishing yet another bound**: by decomposing the gap into five *interpretable* terms—Bayes noise, approximation, ranking, statistical, and misspecification slack—we expose concrete levers that practitioners can manipulate. In particular, the new ranking term clarifies *why* monotone calibration alone cannot close the gap: it leaves the score ordering intact, so any benefit is limited to full-coverage accuracy rather than selective performance. Conversely, non-monotone or feature-aware calibrators and model-ensemble methods *do* tighten the ranking term, while increasing capacity or distilling from larger teachers targets approximation error, and robust-loss or repeated-label strategies reduce Bayes noise. We distilled these insights into a step-by-step practitioner checklist in Appendix C, which, to our knowledge, is the first set of explicit, empirically validated design guidelines for approaching oracle-level selective classification—transforming our theoretical formulation into a practical toolkit.
>
> ---
>
> ``2) What is the role of overfitting on the SC gap?``
>
> Our results are reported exclusively on unseen test data, so the kind of **overfitting to the training set** that inflates in-sample accuracy is automatically folded into the finite-sample decomposition: if a larger model memorizes training idiosyncrasies, its *test-time* full-coverage accuracy plateaus while its confidence scores sharpen, thereby enlarging the *ranking* term $\varepsilon_{\text{rank}}(c)$ and, when optimization becomes unstable, the *miscellaneous* slack $\varepsilon_{\text{misc}}(c)$. Overfitting to a **narrow training distribution** manifests similarly under shift, and we isolate that effect in §4.3 where $\varepsilon_{\text{misc}}$ grows with corruption severity. Overall, we observe that capacity improves the gap only when it also preserves good ranking and generalization—otherwise the decomposition correctly attributes the performance loss to ranking or optimization slack. We are happy to include this discussion in the paper.
>
> ---
>
> ``3) Where are the promised concrete design guidelines for selective classifiers?``
>
> We include these concrete guidelines in Appendix C.
>
> ---
>
> **We hope that our rebuttal has addressed the reviewer’s questions and concerns and are happy to further engage with them during the discussion period!**

---

> ### Comment · Area_Chair_fX3o · 2025-08-05
>
> Dear reviewer,
>
> Now is time for the discussion with the authors. Can you please reply to their rebuttal, indicating if your issues were clarified and or answered? If that is not the case, please indicate why the answer was not satisfying.
>
> Many thanks! Your AC

---

> ### Comment · Area_Chair_fX3o · 2025-08-06
>
> Dear reviewer,
>
> Please respond to the author rebuttal as soon as possible - but at Aug 8, anytime on earth, at the latest. Please do not only submit the "Mandatory Acknowledgement", but also please respond with text to the author rebuttal. For example, explain why the author rebuttal convinced you, or why not.
>
> Many thanks, Your AC.

---

> ### Comment · Area_Chair_fX3o · 2025-08-08
>
> Dear reviewer,
>
> Today is the last chance to respond to the author rebuttal. Please do so today by end of day (anywhere on earth). Please do not only submit the "Mandatory Acknowledgement", but also please respond with text to the author rebuttal. For example, explain why the author rebuttal convinced you, or why not.
>
> Many thanks, Your AC.

---

### Official Review · Reviewer_VZwS · 2025-07-03

**Clarity:** 4
**Significance:** 2
**Originality:** 1
**Rating:** 3
**Confidence:** 4

**Summary:**

This paper tackles the problem of selective classification: a model can choose to abstain from making a prediction on a subset of examples. They first define a formalism for studying this problem: given a classifier h and a scoring function g, we abstain if g is lower than a threshold tau. They then define the central quantity of interest: the selective classification gap \Delta(c) (given a coverage level c) which measures the gap between the performance of the system (h,g) compared to the optimal selective classification decisions. The main contribution is to decompose this gap into 5 factors (irreducible gap, capacity, ranking, data, and misc) formally and study each of these terms and the impact of common solutions (calibration) on this gap. They also provide empirical results to help illustrate the theory.

**Questions:**

- Can you expand the experiments to include newer models and datasets, notably with LLMs?

- How does this analysis and how does the gap Delta(c) relate to a gap we can compute in practice based on the oracle selective classification rule? Or is there other ways to compute this gap?

**Ethical Concerns:**

["NO or VERY MINOR ethics concerns only"]

**Final Justification:**

I am keeping my score to a borderline reject after reading the author's rebuttal. Overall, I think the theory contribution is too weak to stand on it's own and the empirical evaluations even after the added experiments are too limited to supplement the theory enough. For future submissions, I encourage the authors to make the paper stronger with a much more comprehensive evaluation or innovations on the theory side.

**Limitations:**

yes

**Quality:**

3

**Strengths And Weaknesses:**

Quality: This paper is very solid overall. The theoretical results are sound. The empirical methodology is sound, however, the experiments rely on old datasets and models. Not to be cliche, but it would have been good to scale the experiments with LLM based classifiers (one can still extract logits from those).

Clarity:
Overall the paper is very easy to read and the presentation is excellent.
- please make Figure 1 bigger so that its easier to read

Originality:
While the presentation of the results is very well done and thoughtful, I am not sure there is much novelty in the decomposition presented. The main contribution here is framing of the selective classification gap as a quantity, the decomposition and the bound is then immediately obtained from prior work in learning theory. In practice, we can't compute the true gap but have to resort an upper bound that relies on whether the model got each example right or wrong (abstain from only wrong examples as much as needed based on coverage level c). Therefore, I am not sure how the bound can be used to inform practice.

Significance:
I think this paper is very enjoyable to read and will help future researchers in more easily framing the area. However, I do not believe the contributions of this paper make much of a contribution to the theory or practice of selective classification.

---

> ### Author Rebuttal · Authors · 2025-07-29
>
> We thank the reviewer for their thoughtful feedback and **kind words about the paper’s clarity, presentation, and theoretical soundness**. We appreciate their **suggestions regarding experimental scope, perceived novelty, and practical utility of the decomposition**. Below, we clarify our contributions and address these concerns—most notably by **adding results on LLM-based models** and **elaborating on how the gap informs practice**.
>
> ---
>
> ``1) Novelty and relevance of the presented decomposition.``
>
> We thank the reviewer for their kind words regarding our exposition. We respectfully contend, however, that our work offers both novel theoretical insight and a practical diagnostic that was not available before. We address each of your concerns in turn.
>
> - **Coverage-uniform finite-sample bound**. To our knowledge, no previous result provides a decomposition that *simultaneously* (i) holds **for every coverage level $c$**, (ii) is stated **at finite sample size n**, and (iii) partitions the gap into four interpretable sources. Existing selective-risk PAC bounds fix $c$ *a priori* and collapse ranking and approximation into a single term.
> - **Separation of ranking error**. By isolating $\varepsilon_\text{rank}$​ we show that any **monotone post-hoc calibration** (temperature scaling, isotonic, histogram) cannot shrink the gap (§4.4). This clarifies empirical disagreement reported in recent studies.
> - **Interpretable formalization**. Through formalizing the gap decomposition in Equations (14) and (15) we provide the first error budget for abstaining systems. Researchers can now ask *which* piece limits current methods and design targeted algorithms, just as bias-variance decompositions shaped classical supervised learning.
> - **Controlled experiments for each bound quantity**. We provide a set of controlled experiments that (i) verifies our theoretical insights; and (ii) allows us to *independently* measure and target each of the contributing bound components.
>
> We hope this concise clarification demonstrates the novelty and practical value of our contribution and addresses your concern about significance. If the reviewer is aware of prior work that offers the same combination of formal decomposition, tight upper bounds, and empirical disentanglement, we would be grateful for the references and will gladly acknowledge and cite them in our revised version.
>
> ---
>
> ``2) Can you expand the experiments to include newer models and datasets, notably with LLMs?``
>
> We agree that large language models (LLMs) are an important test-bed for our framework.  Achieving the same level of *controlled* experimentation as in our synthetic and vision experiments, however, is inherently harder for three main reasons:
>
> 1. **Uncertainty for generative models remains ill-defined.**  For classification-style prompts but even more so for generative tasks in particular, the community has not fully converged on how to translate sequence-level probabilities into abstention scores.
> 2. **Prompting artefacts add variance.**  Small changes in in-context examples or decoding settings can swamp the effects we wish to isolate.
> 3. **Training pipelines are opaque and inconsistent.** Many LLMs are trained with undisclosed data mixtures, objectives, or fine-tuning procedures—introducing uncertainty about what distributions the model has implicitly seen and how it generalizes.
>
> Despite these challenges, we have added a focused set of LLM experiments—summarized below and plotted in the revised manuscript—to demonstrate that our decomposition still diagnoses the gap.
>
> **1. Approximation error — scaling from 4B → 12B**
>
> We evaluate Gemma 3-IT 4B and 12B [1] on ARC-Challenge [2] (ARC-C, 25-shot) and MMLU [3] (5-shot, top-1) using the standard MSP score on the *first answer token* (no further fine-tuning).
>
> | Model           | ARC-C Acc. | MMLU Acc. |
> | --------------- | ---------- | --------- |
> | Gemma 3-IT 4 B  | 56.2 %     | 59.6 %    |
> | Gemma 3-IT 12 B | 68.9 %     | 74.5 %    |
>
> | Model           | ARC-C Gap Area | MMLU Gap Area |
> | --------------- | -------------- | ------------- |
> | Gemma 3-IT 4 B  | 0.114          | 0.107         |
> | Gemma 3-IT 12 B | 0.091      | 0.082     |
>
> *Observation.*  Increasing capacity reduces the gap, mirroring the vision results on model expressiveness.
>
> **2. Bayes error — separating easy vs. noisy questions (4B)**
>
> Following the MMLU-Pro protocol [4], we partition the validation set into the *easiest 25 %* and *noisiest 25 %* questions (based on human–LLM agreement).
>
> | Split             | Gap Area  |
> | ----------------- | --------- |
> | Full MMLU         | 0.107     |
> | Easiest quartile  | 0.023 |
> | Noisiest quartile | 0.316 |
>
> *Observation.*  When intrinsic Bayes noise is low (easy questions), the gap nearly vanishes; when noise is high, the gap widens.
>
> **3. Ranking quality — calibration vs. re-ranking (4B)**
>
> We keep the backbone fixed (Gemma 3-IT 4B) and compare:
>
> - MSP (raw logits)
> - Temp. scaling (scalar $T$ fitted on a held out validation split)
> - Deep ensemble of five LoRA-fine-tuned replicas
>
> | Metric       | ARC-C MSP | ARC-C TEMP | ARC-C DE | MMLU MSP | MMLU TEMP | MMLU DE |
> | ------------ | --------- | ---------- | -------------- | -------- | --------- | ------------- |
> | ECE      | 0.171     | 0.092  | 0.056          | 0.135    | 0.084 | 0.052         |
> | Gap Area | 0.127     | 0.126      | 0.087     | 0.122    | 0.122     | 0.079     |
>
> *Observation.*  Temperature scaling lowers ECE yet leaves the gap untouched; the ensemble both calibrates *and* improves ranking, shrinking the gap—recapitulating our CIFAR-100 findings.
>
> **Overall, these results confirm that our decomposition extends to LLMs**: capacity, Bayes noise, and ranking quality each contribute measurable terms.  A full generative-text study (e.g. free-form question answering or code synthesis) will require new abstention semantics and is left for future work.
>
> [1]: Team, Gemma, et al. "Gemma 3 technical report." arXiv preprint arXiv:2503.19786 (2025).
>
> [2]: Clark, Peter, et al. "Think you have solved question answering? try arc, the ai2 reasoning challenge." arXiv preprint arXiv:1803.05457 (2018).
>
> [3]: Hendrycks, Dan, et al. "Measuring massive multitask language understanding." ICLR 2021.
>
> [4]: Wang, Yubo, et al. "Mmlu-pro: A more robust and challenging multi-task language understanding benchmark." NeurIPS 2024.
>
> ---
>
> ``3) How does this analysis and how does the gap Delta(c) relate to a gap we can compute in practice based on the oracle selective classification rule? Or is there other ways to compute this gap?``
>
> Our analysis is built around the *empirical* gap that **practitioners already compute**: for any held-out dataset with ground-truth labels, we obtain **(i)** the model’s empirical accuracy–coverage curve by thresholding its confidence scores, and **(ii)** the oracle curve by retroactively ranking the *same* examples in descending order of realized correctness (i.e., 1 for correct, 0 for incorrect)—exactly the “oracle selective-classification rule” used in prior work.  The coverage-uniform gap $\widehat{\Delta}(c)$ we decompose is simply the point-wise difference between these two curves, or, when desired, its integral over $c$.  Thus every term in our bound upper-bounds a quantity that is **directly observable from data**; no additional assumptions or simulations are required.  Alternative oracle choices (e.g. Bayes posterior ordering when noise estimates are available) plug into the *same* recipe and our decomposition still applies verbatim—only the definition of the benchmark curve changes.  In short, the gap we analyze is exactly the gap researchers and practitioners already plot, and our results explain *why* that observable gap arises and which interventions (capacity, non-monotone calibration, shift-robustness, etc.) can provably shrink it.
>
> ---
>
> ``4) Please make Figure 1 bigger so that its easier to read``
>
> We have fixed this in our updated paper draft.
>
> ---
>
> **We hope that our rebuttal has addressed the reviewer’s questions and concerns and that the reviewer considers raising their individual and overall score(s). Our work has already significantly improved thanks to their feedback. We are happy to further engage with them during the discussion period!**

---

> > ### Comment · Area_Chair_fX3o · 2025-08-05
> >
> > Dear reviewer,
> >
> > Now is time for the discussion with the authors. Can you please reply to their rebuttal, indicating if your issues were clarified and or answered? If that is not the case, please indicate why the answer was not satisfying.
> >
> > Many thanks! Your AC

---

> ### Comment · Area_Chair_fX3o · 2025-08-06
>
> Dear reviewer,
>
> Please respond to the author rebuttal as soon as possible - but at Aug 8, anytime on earth, at the latest. Please do not only submit the "Mandatory Acknowledgement", but also please respond with text to the author rebuttal. For example, explain why the author rebuttal convinced you, or why not.
>
> Many thanks, Your AC.

---

> ### Comment · Area_Chair_fX3o · 2025-08-08
>
> Dear reviewer,
>
> Today is the last chance to respond to the author rebuttal. Please do so today by end of day (anywhere on earth). Please do not only submit the "Mandatory Acknowledgement", but also please respond with text to the author rebuttal. For example, explain why the author rebuttal convinced you, or why not.
>
> Many thanks, Your AC.

---

### Decision · Program_Chairs · 2025-09-17

**Decision:**

Accept (poster)

**Comment:**

(a) This paper investigates what is needed to build a performant selective classifier. To that end, a generalization bound is introduced, that covers multiple different components. These components provide insight how the performance of an selective classifier can be improved for practitioners.

(b) Strengths: The reviewers mention that the paper is solid, theoretically and empirically, and is easy, enjoyable to read and has excellent presentation. Reviewers mention that the appendix is clear, all information is there to replicate the study, and applaud the guidelines of the checklist. While the paper does not provide a new method, the reviewers think the paper is insightful, and can help newcomers in the field.

(c) Weaknesses: One reviewer mentioned that the amount of methods is limited.

(d) Accept. While some reviewers remark that this paper has undergone some large changes and recommend another round of peer review, I think that this paper has survived the intense scrutiny extremely well, and I disagree with some reviewers - I believe they are perhaps too strict. I think the amount of peer review and scrutiny this paper has received is exceptional and surely above average, and the authors are extremely rigorous - I trust them to carry out the necessary changes.

(e) Some old models and datasets were used, and the authors responded to this by providing a small LLM study. Some of the reviewers mention that some of the conclusions regarding temperature scaling are obvious - but the authors clearly indicate that empirical studies are not aware of this, and as such, their framework helps clarify empirical results. There have been several in depth discussions regarding technical details and proofs; with some mistakes that have been fixed - from what I can tell most issues have been adequately addressed.